biomechanics, ecology, evolution

fish, locomotion, suction-feeding, hydrodynamics, Reynolds number, Womersley number

**Author for correspondence:**
Karin H. Olsson
e-mail: olsson.karin.h@gmail.com

# Trophic guilds of suction-feeding fishes are distinguished by their characteristic hydrodynamics of swimming and feeding

Karin H. Olsson[1,2], Roi Gurka[3] and Roi Holzman[1,2]

[1]School of Zoology, George S Wise Faculty of Life Sciences, Tel Aviv University, Tel Aviv, Israel
[2]Interuniversity Institute for Marine Sciences in Eilat, Eilat, Israel
[3]Department of Physics and Engineering Science, Coastal Carolina University, Conway, SC, USA

KHO, 0000-0002-1695-0989; RG, 0000-0002-8907-6663

Suction-feeding in fishes is a ubiquitous form of prey capture whose outcome depends both on the movements of the predator and the prey, and on the dynamics of the surrounding fluid, which exerts forces on the two organisms. The inherent complexity of suction-feeding has challenged previous efforts to understand how the feeding strikes are modified when species evolve to feed on different prey types. Here, we use the concept of dynamic similarity, commonly applied to understanding the mechanisms of swimming, flying, walking and aquatic feeding. We characterize the hydrodynamic regimes pertaining to (i) the forward movement of the fish (ram), and (ii) the suction flows for feeding strikes of 71 species of acanthomorph fishes. A discriminant function analysis revealed that feeding strikes of zooplanktivores, generalists and piscivores could be distinguished based on their hydrodynamic regimes. Furthermore, a phylogenetic comparative analysis revealed that there are distinctive hydrodynamic adaptive peaks associated with zooplanktivores, generalists and piscivores. The scaling of dynamic similarity across species, body sizes and feeding guilds in fishes indicates that elementary hydrodynamic principles govern the trophic evolution of suction-feeding in fishes.

## 1. Introduction

Feeding is a complex behaviour in which multiple traits interact to determine its outcome. Successful completion of a feeding task relies on locating the prey, closing the distance to it, and engulfing it, while responding to the prey's defence mechanisms (escape, manoeuvring, etc.). Each of these steps involves the integration of skeletal, muscular and sensory systems of the predator. The inherent complexity that underlies this interaction challenges our attempts to understand the contribution of each trait to feeding success, and the selective pressures that act upon these traits. This complexity is exaggerated in the aquatic realm. In the aquatic realm, the behaviours of the organisms are coupled with the dynamics of the fluid, which exerts forces on the organisms that must be accounted for. For example, in herbivorous fishes that graze on algae attached to the substrate, the thrust generated by the fins is used to dislodge the food from its holdfast, and the speed of reversing away from the substrate determines the amount of algae removed per feeding bout [1]. Conversely, feeding success in larval fishes is constrained by their inability to exert sufficient force on the prey owing to the viscous interaction between the water and the larvae [2,3].

Suction-feeding in fishes has challenged previous efforts to link morphology to diet specialization. A suction-feeding strike combines rapid swimming towards the prey (hereafter 'ram') with the generation of suction flows (hereafter 'suction') that draw the prey towards the mouth [4–6]. Suction-feeding is a ubiquitous and evolutionarily conserved mode of prey capture [4,7], used to capture a wide range of prey, from hard-shelled organisms that

attach tightly to the substrate, to mobile, evasive prey [8–10]. However, despite an increasingly refined understanding of the hydrodynamic mechanisms controlling this process (e.g. [6,11–16]), it remains unclear how this evolutionarily conserved behaviour can be conscripted and modulated to accommodate such a diverse range of prey. Specifically, the contrasting requirements between capturing mobile prey and dislodging attached prey gave rise to the hypothesis that the relative contribution of ram speed and suction should differ for different prey (e.g. [7,17–21]). However, this approach has been challenging to generalize beyond species-pairs comparisons, partly because ram can differ across species by several orders of magnitude, while the suction flows are highly conserved across species and their reach is limited to about one mouth diameter [7,16,20–22].

A useful approach for characterizing complex behaviours is to use the concept of dynamic similarity. In the case of fluid mechanics, dynamic similarity exists between two flow cases (e.g. the flow generated by small and large organisms in fluids of different viscosities [23]) if the forces they experience are parallel, relate in magnitude and scale by a constant factor [24,25]. Hence, dynamic similarity is employed in situations that involve mass in motion. Mathematically, the scaling of the forces is expressed as a ratio, i.e. a dimensionless number, and the nature of the flow determines the appropriate dimensionless number(s) used to assess dynamic similarity. This concept enables a comparison of the hydrodynamics that govern the behaviours of animals of different sizes, speeds and shapes. For example, swimming in fishes is often characterized in terms of the Strouhal number, which provides the ratio of unsteadiness to inertial forces in oscillating flows, and thus links the tail beats to the propulsive efficiency across different sizes and species of fishes [26,27]. More generally, organisms that rely on undulatory motions for their propulsion demonstrate a constant scaling between two dimensionless numbers: the Reynolds and the 'swimming' numbers [28], which denote the ratio between inertial and viscous forces, and tail beat amplitude and frequency to inertial forces, respectively. This scaling reveals that across lengths ranging from a few millimetres to tens of metres, and across the vertebrate phylogeny, basic hydrodynamic principles govern the locomotory dynamics of inertial swimming.

Here, we posit that the hydrodynamics which characterize suction-feeding evolve towards adaptive peaks that are distinctive for different feeding guilds, i.e. that the combination of traits that optimizes suction feeding differs depending on the targeted prey type. Similarly to previous studies [28], we characterize the hydrodynamic regime pertaining to ram using the dimensionless Reynolds number, which couples the length of the fish and its swimming speed (inertial force) and relates these to the viscous force. We follow Krishnan et al. [29] and treat suction feeding flows as a single-pulse event. Accordingly, we characterize the hydrodynamic regime pertaining to it using the dimensionless Womersley number, which is the ratio between the pulsative flow frequency, given by the gape diameter and the angular speed of mouth opening, to the viscous effects [30]. While previous studies [3] used the Reynolds number to characterize suction flows, this is not ideal. The Reynolds number rests on the assumption of steady flow but suction involves abrupt opening and closing of the mouth which generates unsteady flows with steep, temporal velocity gradients [6]. Treating suction as a single-pulse, unsteady flow event is

justified as spatial and temporal gradients (i.e. accelerations) are the dominant source for the hydrodynamic force exerted on attached, swimming and free-floating prey [5,31].

We calculate the Reynolds and Womersley numbers for suction-feeding strikes in 71 species belonging to five radiations of acanthomorph fishes representing three feeding guilds: zooplanktivores, generalists and piscivores. We examine to which extent the different feeding guilds can be distinguished solely by the Reynolds and Womersley numbers, using discriminant function analysis. We employ phylogenetic comparative methods to test the hypothesis that the Reynolds and Womersley numbers evolve towards adaptive peaks that are specific to each feeding guild and quantify the strength of attraction to those peaks.

## 2. Material and methods

### (a) Data acquisition

We obtained suction-feeding kinematics (peak gape, time to peak gape and ram speed) from data collected for three previous publications. These encompassed feeding strikes of 15 species of Centrarchidae [31], 27 species of Serranidae [32] and 21 species of Cichlidae [33]. Original movies from the cichlid study [33] were re-analysed to extract all the kinematic variables relevant to this study (electronic supplementary material, S1 Data collection). These data were supplemented with unpublished data (R. Holzman 2010–2015) on two species of Antennariidae, two species of Centrarchidae, four species of Cichlidae, four species of Pomacentridae and six species of Serranidae.

The Reynolds number Re is calculated as

$$\mathrm{Re} = \frac{uL_1}{\nu} = \frac{\mathrm{ram} \times \mathrm{SL}}{\nu}, \tag{2.1}$$

where $u$ is the ram speed (ram; m s$^{-1}$), and $L_1$ the characteristic length, which corresponds to the standard length (the distance from the snout to the base of the tail fin, SL, m), respectively, of the swimming fish, and $\nu$ (m$^2$ s$^{-1}$) is the kinematic viscosity of the fluid.

The Womersley number $\alpha^2$ is calculated as

$$\alpha^2 = \frac{\omega L_2^2}{\nu} = \frac{2\pi/\mathrm{TTPG} \times \mathrm{PG}^2}{\nu}, \tag{2.2}$$

where $\omega$ is the angular frequency of the pulse, and $L_2$ a characteristic length associated with $\omega$. Following [29], we consider suction as a single-pulse event, in which the angular frequency is the time it takes for the fish to fully open its mouth (time to peak gape, TTPG, s), $\omega = 2\pi/\mathrm{TTPG}$, and the characteristic length is the peak gape diameter (PG, m). For freshwater species (cichlids and centrarchids), we set $\nu = 1.0034 \times 10^{-6}$ m$^2$ s$^{-1}$ (freshwater at 20°C), and for marine species (antennarids, serranids and pomacentrids), we set $\nu = 1.0508 \times 10^{-6}$ m$^2$ s$^{-1}$ (seawater at 20°C).

When using dimensionless numbers, care must be taken to use the parameters (e.g. length and speed) relevant to the hydrodynamic properties of the system investigated. For example, Re can be calculated for different systems such as the body (in which case the characteristic length and speed are SL and ram) or for an appendage (in which case the length and speed of the appendage should be used; [23,34]). According to this principle, $\alpha^2$ characterizes the hydrodynamics of a pulsating cavity, and its behaviour is governed by the aperture diameter and pulsation frequency of the cavity and is unrelated to the scale of the enclosing body (i.e. SL). Consequently, different characteristic lengths ($L_1$ and $L_2$) are required to calculate Re and $\alpha^2$.

We used primary literature (stomach content analysis) to identify the predominant food type and classified each species

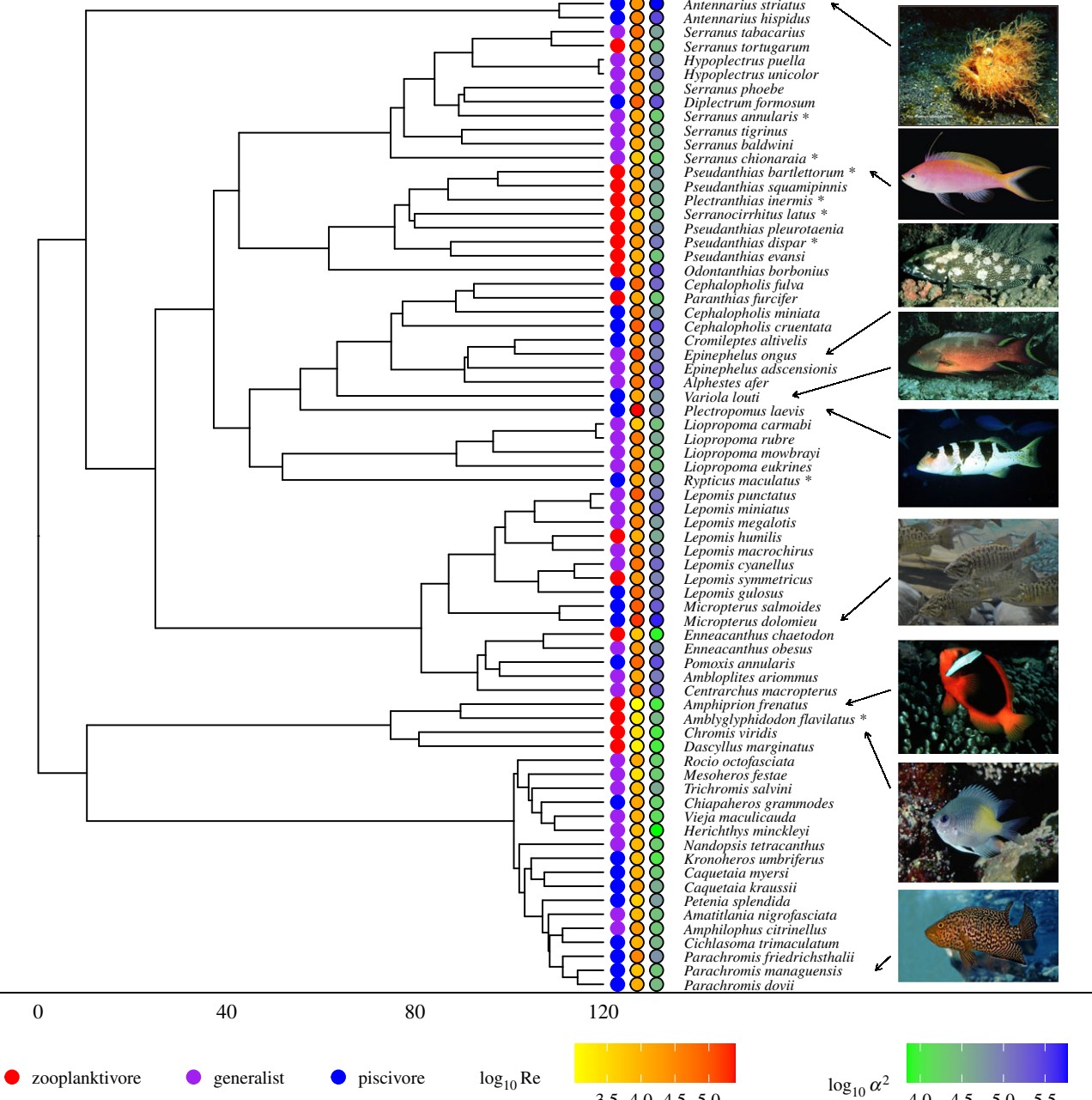

**Figure 1.** Phylogenetic relationships between the 71 species comprising the dataset. Species marked by (*) are positioned randomly within the smallest clade to which they are known to belong (see text). Feeding guild, $\log_{10}$Re and $\log_{10}\alpha^2$ denoted by coloured circles. Images from FishBase.org. (Online version in colour.)

into one of six feeding guilds (detritivore, herbivore, molluskivore, generalist, piscivore and zooplanktivore) and verified the assigned feeding guild using dissimilarity analysis (electronic supplementary material, S2, figure SI 1 and table SI 1). Distinguishing between marine and freshwater generalists did not affect the results (electronic supplementary material, 4.2, tables SI 3, 4 and figure SI 3). Owing to low species representation in the detritivore, herbivore and molluskivore guilds (2–4 species each), species from these guilds were excluded from further analyses. Thus, the final dataset consists of the kinematics characteristics of 979 feeding strikes from 154 individual fishes, representing 71 species from five families (two Antennariidae, 15 Centrarchidae, 17 Cichlidae, four Pomacentridae and 33 Serranidae; figure 1).

For each strike, we calculated Re and $\alpha^2$ using equations (2.1) and (2.2). For 64 of the 71 species, standard length was reported by the authors and for the remainder we obtained length measurements from FishBase (fishbase.org, [35]). Excluding these seven species from the analysis yielded results that were consistent with those of the entire dataset (electronic supplementary material, S4.2, table SI 3 and figure SI 3). To avoid biasing the results towards species for which there were many data points, we reduced the dataset to the species average by first calculating the mean Re and $\alpha^2$ for each individual and then the mean for each species. Both Re and $\alpha^2$ were $\log_{10}$-transformed to obtain normal distributions, verified using the Shapiro–Wilks test for normality ($p < 0.05$ for non-transformed data and $p > 0.1$ for the log-transformed Re and $\alpha^2$).

To determine the extent to which Re and $\alpha^2$ could be associated with a feeding guild, we performed a discriminant function analysis. Standard linear discriminant analysis assumes that each class (hereafter the three feeding guilds) has a single Gaussian distribution, while mixture discriminant analysis relaxes this assumption and allows each class to be a Gaussian mixture of subclasses. We performed both analyses and found that the mixture discriminant analysis performed better; therefore, we only report results from the latter. This method cannot account for phylogenetic relationships and was thus performed on phylogenetically uncorrected data.

## (b) Phylogenetic methods

We hypothesized that fishes from different feeding guilds evolve towards different adaptive peaks, which are defined by their distinctive hydrodynamic regimes. The location of an extant species with respect to an adaptive peak, however, is dependent on the strength of the selective forces that operate on it and its phylogenetic history. For example, the transition of a species that diverged from a piscivorous ancestor (i.e. residing near the adaptive peak for piscivores) to planktivory may take more time to reach the adaptive peak for planktivores than a species that diverged from a generalist ancestor. To account for such phylogenetic effects, we compared several models of trait evolution, pertaining to effect of the feeding guilds on the tempo (rate) and mode (direction) of evolution. These models estimate, based on the phylogeny and the distribution of traits in the extant species, the likelihood that each feeding guild is characterized by a different rate and a different direction of trait evolution. The support for the hypothesis can subsequently be tested using model selection.

In general, traits are considered to evolve in response to a combination of drift and selection. Drift results from inherent variation within a population, while selection is owing to differential fitness for different phenotypes. Statistically, this can be modelled using an Ornstein–Uhlenbeck (OU) model:

$$dX_{(t)} = \beta(\theta - X_{(t)}) + \sigma dB_{(t)}, \tag{2.3}$$

where $X_{(t)}$ is the value of trait $X$ at time $t$, $\theta$ the optimal trait value (also known as the adaptive peak), $\beta$ the pull towards the optimum, $dB_{(t)}$ the Brownian motion (BM) at time $t$, and $\sigma$ is the rate of the BM. The BM is the non-directional divergence of the trait when the evolution is only affected by drift. Thus, if there is no selection operating on the trait, the first term on the right-hand side of equation (2.3) collapses to 0, and the model is known as a BM model, in which the average trait divergence between two species is proportional to the evolutionary time separating the species and the rate of the BM. The full OU model can estimate the likelihood of state-dependent parameters, such as the existence of different optima ($\theta$) for each trophic guild. These optima can be thought of as the trait value to which all species from the appropriate guild will arrive given sufficient time to evolve. By allowing $\sigma$ to be common to all guilds or specific to each guild, and $\theta$ to be either 0 common to all guilds or specific to each guild, a set of nested BM and OU models can be specified (two BM models and four OU models, see table 1 for each model specification). Importantly, three models (BM1, BMS and OU1) specify no optimum or an optimum common to all guilds and thus do not support the hypothesis that Re and $\alpha^2$ are affected by feeding guild, whereas the remaining three (OUM, OUMA and OUMV) all specify guild-specific optima and thus support the hypothesis.

We obtained phylogenetic relationships from a large phylogeny of approximately 11 000 species of fishes, based on multigene molecular data analyses and time-calibrated using fossil records ([36,37]; electronic supplementary material, S3). The evolutionary history of the three feeding guilds in our dataset is not known; therefore, it was reconstructed using stochastic character mapping (make.simmap function from the phytools package, [38,39]). This allows trait history to be stochastically simulated based on the states at the tips of the tree in order to generate hypotheses pertaining to the evolution of each feeding guild and the transitions between them. For each of the 20 randomly generated phylogenies, we ran make.simmap 50 times to produce a total of 1000 state-mapped trees.

The extent to which phylogeny affects the evolution of a trait (the phylogenetic signal) can be quantified using Pagel's $\lambda$ [40,41], which ranges from 0 where there is no effect of phylogeny to 1 in the case of pure Brownian motion. To estimate the effect of phylogeny on the evolution of Re and $\alpha^2$ we calculated

**Table 1.** Illustration of parameters fitted in common to all feeding guilds (light grey), specific to each feeding guild (dark grey), or not included in the model (white) for the different BM and OU models. (Colours indicate single-optimum (red) and guild-specific optima (green) models. Two BM models were fitted: BM1 in which the BM rate is modelled as common to all guilds, and BMS, in which the BM rate is modelled as specific to each guild. Four OU models were fitted: (i) OU1 in which fishes are modelled to evolve towards a common optimum at the same BM rate with the same pull; (ii) OUM in which fishes from different guilds are modelled to evolve towards different optima, but at the same BM rate and the same pull; (iii) OUMA in which fishes from different guilds are modelled to evolve towards different optima with different pulls, but at the same BM rate; and (iv) OUMV in which fishes from different guilds are modelled to evolve towards different optima, at different BM rates, but with the same pull.)

| model | | model parameter | | |
| --- | --- | --- | --- | --- |
| | | optimum | pull | rate |
| | OUMV | S | C | S |
| | OUMA | S | S | C |
| | OUM | S | C | C |
| | OU1 | C | C | C |
| | BMS | 0 | 0 | S |
| | BM1 | 0 | 0 | C |

Pagel's $\lambda$ using the phylosig function from the phytools package [39] for both traits across the 20 randomly generated trees.

To analyse how the evolution of Re and $\alpha^2$ are affected by the feeding guild, we modelled the evolution of each trait by fitting the BM and OU models using the OUwie package [42] for each tree. For each model in each tree, we verified that the maximum likelihood estimate was reliable by performing an eigen decomposition of the Hessian matrix and confirming that it was positive-definite. Models for which this was not the case were excluded. We obtained the estimated trait optimum for each tree using model averaging by extracting the Akaike information criteria with small sample correction (AICc) and the guild-specific optimal trait value [43]. For the BM models (BM1 and BMS), optimum was set to 0. We calculated the Akaike weights from the AICc of the models and used these to calculate a weighted estimate of the optimum. We repeated this procedure for the estimate of the pull. For each trait (Re and $\alpha^2$), we also estimated the overall support for the different models by comparing the distribution of the Akaike weights (range 0–1) across all trees. Models that are highly supported have Akaike weights close to 1, while models that have poor support have weights close to 0. This procedure yielded 1000 estimates for the optimal value of Re and $\alpha^2$, and of the strength of the pull towards it. To visualize the joint distribution of the estimates for optimal Re and $\alpha^2$, we performed separate kernel density estimates for each guild and computed the contours describing the areas which comprised 25%, 50% and 75% of the estimated optima.

Organismal size is known to be a key variable affecting species ecology. Re and $\alpha^2$ are calculated using length, speed and viscosity. While marine and freshwater species experience slightly different viscosities, there is a 10-fold difference in lengths (range SL: 28.5–307.7 mm, range PG: 3.27–29.8 mm), necessarily producing a relationship between fish size and the hydrodynamic regime of its movements. Our approach was to ask if size alone is an equally good explanatory variable as the hydrodynamic regime; or whether the hydrodynamic regime

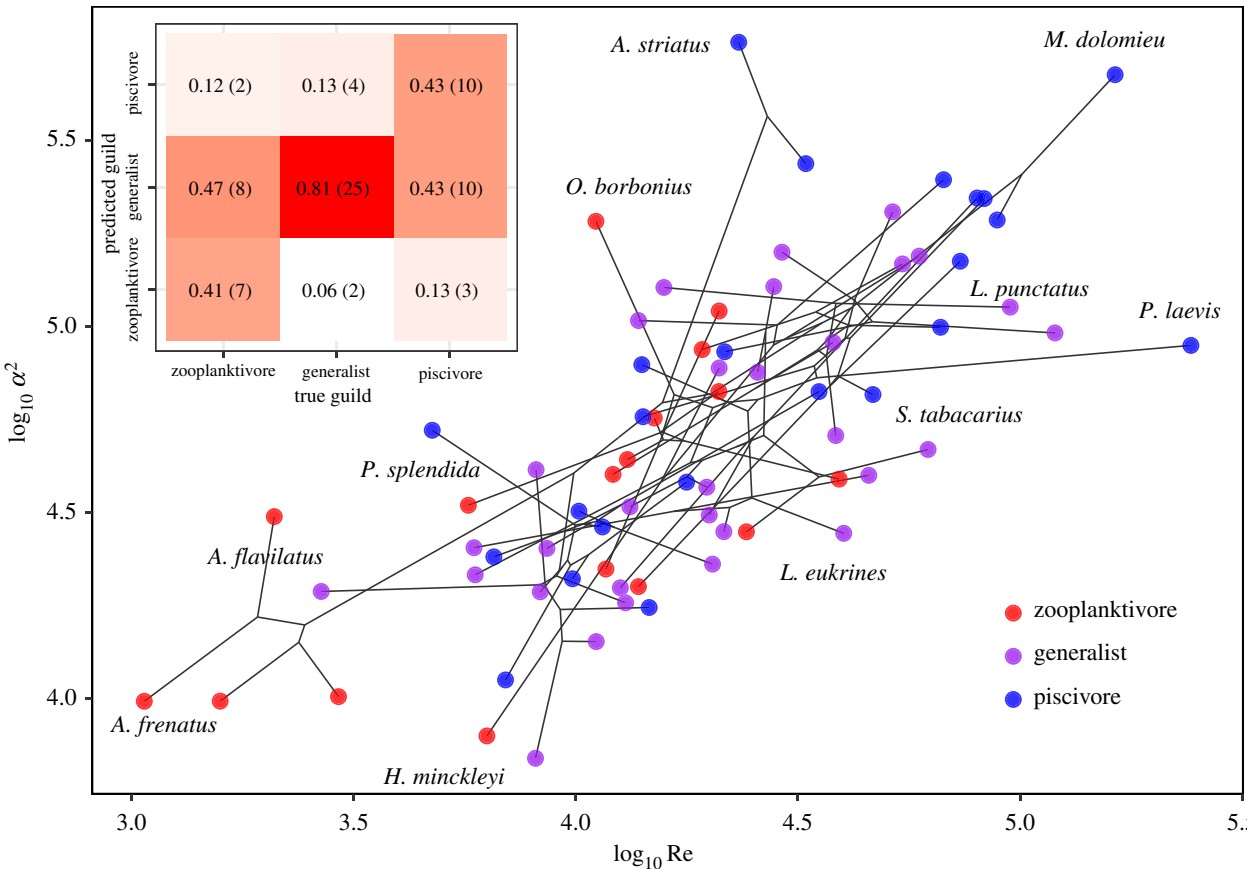

**Figure 2.** Phylomorphospace (projection of the phylogenetic tree into trait space with branches shown in grey lines) for Re and $\alpha^2$ of the species in the dataset. Each point represents a species with colour denoting zooplanktivores, generalists and piscivores. For clarity, only a subset of points has been labelled (all species data can be found in the electronic supplementary material, table SI 1). Upper left inset presents the confusion table for the mixture discriminant analysis. Labels denote proportion (number) of species classified into each guild. Branches crossing over each other, and the disordered appearance of the underlying phylogenetic tree, are consistent with traits evolving towards guild-specific optima [45]. (Online version in colour.)

conveys more information about the evolution of feeding guilds. We tested this by fitting the phylogenetic models (table 1) to Re and SL, and to $\alpha^2$ and PG for the 1000 state-mapped trees, and (i) calculated the median AICc for each model, and (ii) for each tree identified the model that in terms of AICc best fitted the evolution of Re or $\alpha^2$, and compared it to an identically specified model with SL or PG. In addition, we repeated the analysis on size-corrected ram and gape speeds (now in units of SL s$^{-1}$). A general point to make here is that while body size indeed affects numerous other processes, the characterization of the hydrodynamic regime ubiquity makes a specific, functional link between body size and feeding ecology.

All analyses were performed in R v. 4.0.3 [44].

## 3. Results

### (a) Discriminant analysis

The species in our dataset are approximately distributed along a slope from low to high values of Re and $\alpha^2$ (figure 2). Combinations of low Re and low $\alpha^2$ (lower left corner in figure 2) are dominated by zooplanktivores, whereas combinations of high Re and high $\alpha^2$ (upper right corner in figure 2) are dominated by piscivores. Accordingly, mixture discriminant analysis showed that the combination of Re and $\alpha^2$ provides a good predictor of the feeding guild with a mean classification accuracy of 55.1%. For zooplanktivores, the accuracy was 41.2%, while for piscivores, it was 43.5% and for generalists 80.6%. Misclassification was mostly that of zooplanktivores or piscivores classified as generalists (47.1% and 43.5%, respectively). Based

on Re and $\alpha^2$, piscivores were rarely misclassified as zooplanktivores (13%), and vice versa (11.8%).

### (b) Phylogenetic analysis

Trait evolution is affected by both drift and selection, and the opportunity for selection depends on evolutionary time. To assess the effect of phylogeny on the distribution of Re and $\alpha^2$, we calculated the phylogenetic signal. The phylogenetic signal was strong with the mean (±s.e.) Pagel's $\lambda$ for Re being $\lambda = 0.644$ (±0.00418) and for $\alpha^2$ being $\lambda = 0.834$ (±0.0116) (the signal was significant with $p < 0.001$ for all 20 phylogenetic trees used). A phylogenetic generalized linear model showed a significant correlation between Re and $\alpha^2$ following the scaling $\log_{10}\alpha^2 = 0.53\log_{10}R + 2.51$ ($R^2 = 0.31$; $F_{1,69} = 33.07$; $p < 0.001$; model $\lambda$ set by maximum likelihood).

A comparison of trait-evolution models supported the hypothesis that Re and $\alpha^2$ evolve towards distinct, guild-specific adaptive peaks. For Re and $\alpha^2$, the multi-optima models (especially OUM and OUMA, and to a lesser extent also OUMV) received the highest support with inter-quartile range for the Akaike weights for Re ranging from 0.071 to 0.829 and for $\alpha^2$ from 0.0414 to 0.648 (figure 3a). For both Re and $\alpha^2$, the no-optimum BM and the single-optimum OU models (BMS, BM1 and OU1) received the lowest support (inter-quartile range for the Akaike weights for Re range 0.0000004–0.00374, and $\alpha^2$ 0.00588–0.0833).

To assess the strength of the attraction to the trait optimum, we compared the distributions of the estimated pull

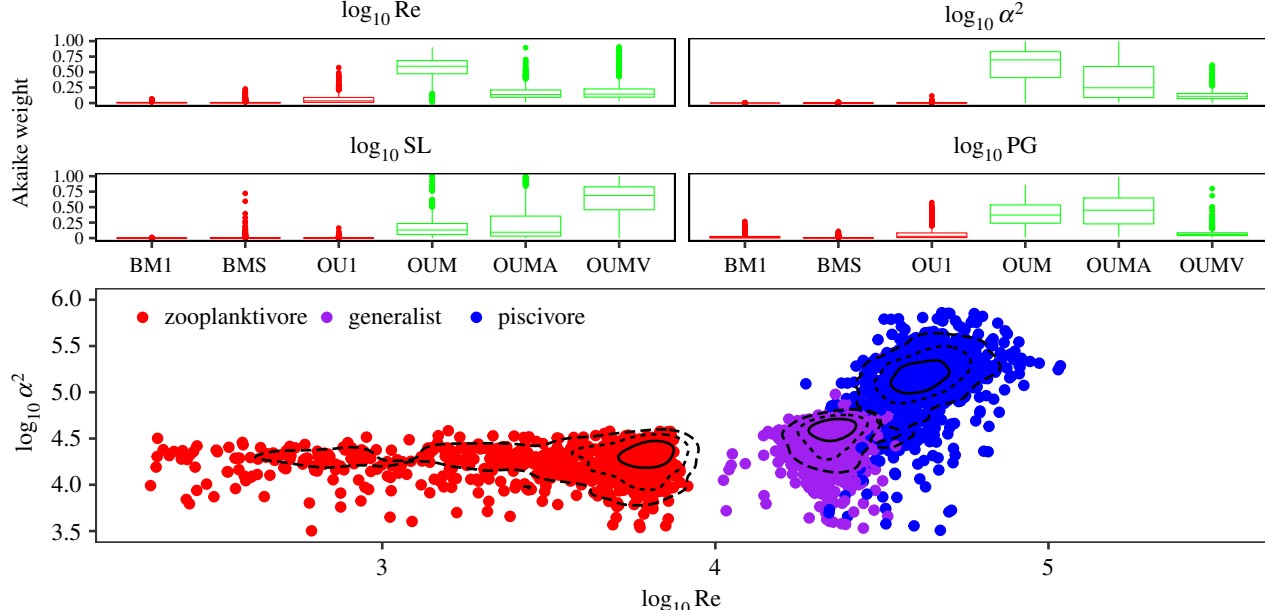

**Figure 3.** Phylogenetic comparative models support the existence of multiple adaptive peaks for the three feeding guilds: (a) model support based on Akaike weights across all trees for Re and $\alpha^2$; colours indicate single-optimum (red) and multi-optima (green) models, and (b) the distribution of model-averaged estimated trait optima for each guild, obtained for 1000 state-mapped trees. Contour lines demarcate 25% (solid line), 50% (dotted line) and 75% (dashed line) of the distributions. Contour lines are calculated using a smoothing parameter, resulting in a slight visual mismatch between the left-skewed and truncated distribution of Re for zooplanktivores and the contour lines). (Online version in colour.)

Proc. R. Soc. B 289: 20211968

for each guild and trait. For Re, the distribution of the estimated pull towards the trait optimum differed little between generalists and piscivores (inter-quartile range generalists: 0.043–0.056, piscivores: 0.041–0.056), while for zooplanktivores, the distribution was wider and extended towards lower pull estimates (inter-quartile range zooplanktivores: 0.033–0.055). For $\alpha^2$, the pull was considerably stronger (somewhat overlapping) for zooplanktivores than for the other two guilds (inter-quartile range zooplanktivores: 0.018–0.023, generalists: 0.013–0.019, piscivores: 0.014–0.02).

For zooplanktivores, the estimated optima are distributed along with a wide range of low to medium values of $\log_{10}R$, but are concentrated in a narrow band around $\log_{10}\alpha^2 \approx 4.24$ (figure 3b). For piscivores, the corresponding distribution of estimated optima enclose a region of high Re and high $\alpha^2$, while the estimated trait optima for generalists are characterized by slightly higher $\alpha^2$ compared to zooplanktivores and slightly lower Re compared to piscivores. The contour lines demarcating 25% and 50% of the distribution of the trait optima distinguish three fully separated regions associated with each feeding guild. The 75% contour line results in some overlap between the regions occupied by piscivores and generalists, respectively, while the region occupied by zooplanktivores remains fully separated from those of the other two feeding guilds. Therefore, even after accounting for the uncertainty in the reconstruction of species history, the adaptive peaks for each guild are distinctive.

Examining figure 2 separately for each family show that among the families: centrarchids and cichlids, piscivores generally have higher Re and $\alpha^2$ than the generalists and the centrarchid zooplanktivores the lowest Re and $\alpha^2$, while the patterns among serranids is more complicated (electronic supplementary material, figure SI 4). The two antennarid species are situated at the upper edge of the optimal trait region of piscivores, while the four zooplanktivorous pomacentrids occupy the lower left edge of the optimal trait region for zooplanktivores. In general, the crossing over of branches which leads to a disordered appearance of the underlying phylogenetic tree is consistent with traits evolving towards different optima in response to a shift in the feeding guild of the lineage [45].

To assess if hydrodynamic traits are superior to length-based traits as predictors of the evolution of suction-feeding in fishes, we first compared the support for hydrodynamic models (i.e. pertaining to Re and $\alpha^2$ to that of the length-based models (pertaining to SL and PG, respectively) using the median AICc for all six models. For all traits, the multi-optima models were favoured and median AICc for the best supported hydrodynamic model (OUM for Re and OUMA for $\alpha^2$) was lower than all those for the length-based models (electronic supplementary material, table SI 2). Second, we compared the AICc value of the best model for the evolution of the hydrodynamic traits (Re and $\alpha^2$) in each of the 1000 state-mapped trees to that of the identically specified model for the respective length trait (SL and PG). We found that the AICc was lower for models pertaining to the hydrodynamic traits (median difference = −11.29 for Re-SL and −54.08 for $\alpha^2$-PG, electronic supplementary material, figure SI 2). The lower AICc scores of models pertaining to hydrodynamic traits support the inference that this is a better predictor of the evolution of suction-feeding fishes than size alone. Additionally, distinctive peaks for the three trophic groups were identified for size-corrected ram and gape speed (electronic supplementary material, tables SI2 and SI4), further supporting the conclusion that the observed effects are not caused solely due to body size.

## 4. Discussion

We have shown here that the hydrodynamic regime which pertains to the contribution of swimming (given by the Reynolds number), and the hydrodynamic regime that pertains

to suction (given by the Womersley number) clarify much of the ecological variation in suction-feeding fishes (figure 2). A phylogenetic comparative analysis revealed that these two regimes have evolved towards distinctive adaptive peaks for zooplanktivores, generalists and piscivores (figure 3). This pattern is consistent across families (electronic supplementary material, figure SI4). The scaling of the Reynolds and Womersley numbers across species, body sizes and feeding guilds in fishes indicates that elementary hydrodynamic principles govern the trophic evolution of suction-feeding in fishes. These results demonstrate that combining engineering principles with comparative phylogenetic methods can be a powerful tool to understand the evolution of complex organismal behaviours.

The selection of Reynolds and Womersley numbers in different feeding guilds results in distinctive hydrodynamic regimes for zooplanktivores, generalists and piscivores. High Reynolds numbers are driven by high ram speed and large body size, which is consistent with attacking prey like fishes, which primarily rely on vision to detect a predator and are capable of evasion by a fast burst of swimming [10,46–48]. Conversely, many zooplankton are capable of sensing hydrodynamic disturbances through the flex of hairs on their antennae [49–51]. In the case of fishes, the magnitude of this disturbance generally increases with the cross-sectional area of the fish and its swimming speed through the water [52–54]. In addition to the low Reynolds numbers associated with zooplanktivory in our analysis, previous work on the evolution of zooplanktivory has identified a concomitant reduction in facial features (including jaw length, premaxilla length, distance between the eye, the base of the pectoral fin and the anterior tip of the dentary of the jaw, and the adductor muscle weight; [55]), and the anterior length and region (demarcated by tip of the premaxilla, anterior orbit margin and the articular-quadrate lower jaw joint; [56]), which presumably may contribute to a smaller cross-sectional area. By contrast, high Womersley numbers are driven by large gape size, which corresponds to high flow speed [16]. In our dataset, the highest Womersley numbers were calculated for the antennarids, which are specialized ambush-feeders that depend on very high flow speeds, rather than a fast approach [22]. Optimal Womersley numbers for zooplanktivores span a strikingly narrow range and are also subject to a strong pull, reflecting a strong selection on small gape sizes. This may be consistent with the general reduction in facial features associated with zooplanktivores but may also correspond to a size sufficient for feeding on small prey. The generalist feeding guild in our dataset comprises a diverse set of species from three families (cichlids, centrarchids and serranids) with different diet compositions. In particular, while freshwater generalists frequently incorporate insects and insect larvae in their diets [57], marine generalists often feed on crustaceans [58]. Nevertheless, separating the marine and freshwater generalist guilds their respective optimal trait regions characterized by slightly lower Reynolds numbers than those of piscivores and slightly higher Womersley numbers than those of zooplanktivores, overlapped considerably (electronic supplementary material, figure SI 3). This presumably reflects a trade-off between capture performance of a variety of different prey types.

The phylogenetic signal calculated for the Reynolds and Womersley numbers indicates that both traits are evolutionarily conserved, and the positions that species occupy within the space described by the hydrodynamic regimes of swimming and suction are reflected by both feeding guild and the degree of feeding specialization within each radiation. Antennarids are specialized ambush predators [59], characterized by high Womersley numbers and relatively high Reynolds numbers. Conversely, the pomacentrids encompass several genera of exclusive or partial planktivores [60], and the four pomacentrid species in our dataset occupy positions of low Reynolds and Womersley numbers. Both the centrarchids and the cichlids are multi-guild genera and occupy regions approximately overlapping the generalist optimum. The serranids, which also comprise representatives of all three feeding guilds, deviate from this pattern. Unlike the centrarchids, for which the phylogeny is quite well resolved [61] and the neotropical heroine cichlids, for which advances are made [33,62], the phylogeny of Serranidae is notoriously challenging. This is owing to a large number of taxa, occasionally inadequate species descriptions, and a circumtropical distribution of members of the family [63–65].

This study focused on the hydrodynamic regimes pertaining to swimming and suction. Many suction-feeding fishes are capable of considerable forward extension of their jaws, thus closing the distance to the prey more swiftly than ram alone would allow [22,66–68]. This is an important aspect of prey capture success, but in the absence of an appropriate dimensionless number to describe the hydrodynamics of jaw protrusion, it is not dealt with here. We conclude that in lieu of comparative morphological analyses prey capture can be characterized as an interaction between solid structures, in the form of the bodies of the fish and the prey, and the fluid flow dynamics. The commonly used Reynolds number is often employed to characterize locomotion in aquatic organisms but it does not account for the highly unsteady force component associated with suction. Instead, the temporal aspect of suction-feeding can be captured by the Womersley number, which provides an informative contribution to the analysis of suction-feeding strikes and in combination with the Reynolds number captures a considerable amount of the ecological variation in suction-feeding fishes.

The general approach presented here can be used whenever the forces that govern the behaviour of organisms are understood. For example, the flight performance of wings in birds and bats can be characterized using the lift to drag ratio [69]. Similarly, the forces that characterize walking on water can be expressed using three dimensionless numbers (Bond, Reynolds and Weber; [70]). The Froude number was used to characterize terrestrial bipedal and quadriplegic locomotion [71]. The framework used here enables testing whether these non-dimensional numbers evolve with respect to ecological axis of interest such as trophic ecology, migration distance or habitat type (e.g. forest, desert and montane). This is specifically useful if the relationship between the forces and the ecological axis hypothesized *a priori*.

Data accessibility. Data are available from the OSF repository: https://osf.io/36r2p/ and from the Dryad Digital Repository: https://doi.org/10.1056/dryad.djh9w0w1z [72].

Authors' contributions. K.H.O.: data curation, formal analysis, funding acquisition, investigation, methodology, validation, visualization, writing—original draft and writing—review and editing; R.G.: conceptualization, methodology and writing—review and editing;

R.H.: conceptualization, data curation, funding acquisition, methodology, project administration, resources, supervision and writing—review and editing.

All authors gave final approval for publication and agreed to be held accountable for the work performed therein.

Competing interests. We declare we have no competing interests.

Funding. The research was supported by the U.S.- Israel Binational Science Foundation (BSF) to R.H. (grant no. 2016136). K.H.O. was supported by an IUI Post-Doctoral Fellowship Award.

Acknowledgements. We thank the members of the Holzman lab for valuable discussions, and the IUI for the use of facilities, financial support and encouragement.

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
