## [Peer Review File · Proceedings of the Royal Society B: Biological Sciences]

Review History

RSPB-2021-1383.R0 (Original submission)

Review form: Reviewer 1

Recommendation

Major revision is needed (please make suggestions in comments)

Scientific importance: Is the manuscript an original and important contribution to its field?

Excellent

General interest: Is the paper of sufficient general interest?

Good

Quality of the paper: Is the overall quality of the paper suitable?

Good

Is the length of the paper justified?

Yes

Should the paper be seen by a specialist statistical reviewer?

No

Do you have any concerns about statistical analyses in this paper? If so, please specify them explicitly in your report.

No

It is a condition of publication that authors make their supporting data, code and materials available - either as supplementary material or hosted in an external repository. Please rate, if applicable, the supporting data on the following criteria.

Is it accessible?

Yes

Is it clear?

Yes

Is it adequate?

Yes

Do you have any ethical concerns with this paper?

No

Comments to the Author

This is a terrifically interesting manuscript that introduces the use of a pair of dimensionless ratios to describe the mechanical environment of fish feeding strikes, and then uses these to explore the power of the ratios to describe three trophic groups. The result is a strong pattern of separation of trophic group in the space of the two ratios. The paper has great potential because of its novel use of these two ratios to gain insight into fish feeding diversity. However there are two major issues that have left me less than enthusiastic about the present version of this paper. First, I was confused by some of the categorization of species by diet category. When I looked up several species that I questioned I found data that supported a different category than used in this manuscript. Citations of primary literature are needed for every species – the studies that documented what the fish eat not just books and reviews. Second, my impression is that body size is positively correlated with each of the dimensionless ratios and it is well known that piscivores tend to be large compared to invertivores. So it seems to me that body size may be a trivial explanation for the empirical results presented here. This is the most important issue that needs to be addressed.

L31. The word ‘acanthomorph’ should not be capitalized. Capitalize it if you use the formal group name – Acanthomorpha, but not for the informal name acanthomorph.

L48. I don’t understand this last phrase of the sentence – that thrust from the fins ‘determines feeding rate’? It seems as though you are saying that the amount of thrust that the fins generate limits the feeding rate, but that does not seem very likely. This either needs to be clarified or simplified.

L53. You place the term ‘similitude’ in parentheses here, as another way of saying dynamic similarity, but then this word never appears again in the paper. Perhaps you could just drop it here?

L53-55. The grammar of this sentence is awkward. It reads that you are saying that dynamic similarity can be applied as a dimensionless ratio? Surely that’s not what you intend?

L56. It is not clear what dimensionless numbers you are referring to here. This section needs to be rewritten.

L67. Has suction feeding been difficult to ‘unravel’? Or do you mean that it has resisted attempts to understand the role of dynamic similarity in shaping patterns of its diversity?

L95. Omnivore refer to an animal that eats both plant and animal prey. Is this really what you mean? This is not a ‘feeding guild’ on the same scale as ‘piscivore’ or ‘zooplanktivore’.

L102-. I am concerned about the placement of species into diet categories. There are several issues. First, what do the authors mean by ‘omnivore’. This term refers to an animal that eats animals and plants, but there are many ‘omnivores’ in this study that do not eat plants. Second, exactly how was diet category determined? You cannot just use what different authors say because they may each have a different definition of a piscivore. I looked up half a dozen species from the study that I questioned and I could not find data that supported their categorization in this study (*Thorichthys meeki*, *Trichromis salvini*, *Lepomis humilis*, *Lepomis gulosus*, *Lepomis symmetricus*, *Cichlasoma trimaculatum*, *Plectranthias inermis*). So exactly how were fish categorized for this study? What criteria makes a species piscivorous? Zooplanktivorous? And what do you mean ‘omnivore’? If that is simply a species that is neither a planktivore or a piscivore then it is such a huge, ecological diverse category that I find it very hard to justify.

L104. The Hulsey et al. 2010 paper used for cichlid data does not have any kinematic timing data in it so it is unclear what is meant here that the data used in the present study was taken from that study?

L142. Is it not important to have length data from the individual fish used for timing data etc? Using length data from fishbase is likely to only be accurate to within an order of magnitude. Lab work is normally done on small specimens and adults in the wild are almost always larger than the typical lab experimental fish. This needs to be justified much more fully.

L150. ‘...to one measure for each species...’ This is an average – not a ‘measure’.

L214. Figures 3 & 4. I am concerned that body size is an awkward complication in both the diet data and the dimensionless ratios. On average we would expect larger fish to have higher Reynolds numbers. What is the relationship between Re and body size of the specimens used in the videos?

L214. In a similar way, it appears that the Womersley number calculated here would be positively correlated with body size of the specimens filmed?

L214. And finally, piscivores tend to be larger than invertivores. So, is body size a trivial explanation for the distinction that Womersley number and Re produce between invertivores and piscivores?

Figure 3. There are some things that I find confusing when I compare figure 3 and figure 4. The point for *Lepomis punctatus* in fig 4 (a species mean) is positioned outside the range of all observations on omnivores in figure 3. How can that be. The same is true for the piscivorous cichlid *Petenia splendida* and many other piscivorous cichlids. Species means that fall outside the range of individual observations for that trophic group?? I assume that this has something to do with this being a phylomorphospace but honestly it just does not make sense.

Figure 4. This is a very helpful figure but about half of the species labels are vague about which data point they are meant to label, and many are positioned awkwardly on top of the outlines of the peaks from the OU analysis. This absolutely needs to be cleaned up. Ideally I also recommend that you label more species. I also suggest that the phylogeny in this figure adds very little, so it might help to remove it and possible use lines to connect some points to the names.

Review form: Reviewer 2

Recommendation

Major revision is needed (please make suggestions in comments)

Scientific importance: Is the manuscript an original and important contribution to its field?

Good

General interest: Is the paper of sufficient general interest?

Good

Quality of the paper: Is the overall quality of the paper suitable?

Marginal

Is the length of the paper justified?

Yes

Should the paper be seen by a specialist statistical reviewer?

No

Do you have any concerns about statistical analyses in this paper? If so, please specify them explicitly in your report.

Yes

It is a condition of publication that authors make their supporting data, code and materials available - either as supplementary material or hosted in an external repository. Please rate, if applicable, the supporting data on the following criteria.

Is it accessible?

Yes

Is it clear?

No

Is it adequate?

Yes

Do you have any ethical concerns with this paper?

No

Comments to the Author

Ollson et al have performed a meta-analysis of fish feeding and swimming in the interest of understanding how species from different guilds have evolved with respect to the hydrodynamics of feeding and swimming. I am excited by the premise of this study and very-much like the idea of exploring patterns of diversity with respect to non-dimensional constants from the physical sciences. I think this subject would interest the readers of Proc B and the novel perspective on understanding fish diversity has the potential for a large impact on an area that interests many organismal biologists.

However, I have the following major concerns about this paper.

1. A major component of the analysis of this study generates "trait optima" that are not explained well enough for readers like me to understand their basis. This is arguably the most important feature of the study and it is mysterious to me how it came about and what it means.

2. It seems possible that the results are merely the consequence of the size differences among species. Re and Womersly number (WN) depend on linear measurements of size. The body length (which factors into Re) is probably well-correlated with the gape diameter (which goes into WN). In contrast, the authors discuss Re and WN as if they are independent metrics of swimming and feeding. Since the prey for a species also depends on size, then it would appear possible that size is what predicts guild, Re, and WN, above all other factors. If I am wrong about this, then the paper needs to articulate why that interpretation (which many readers will share) is incorrect.

3. As itemized below, there are many areas of the writing in need of clear definitions and explanations of the justification and/or significance.

SPECIFIC COMMENTS

L27 - I'm not sure what "transition in trophic guilds" means. If "guild" refers to a group of species that exploit a particular niche, then how does a species "transition in" one of those? Please expand or rephrase.

L28 - It would be more valuable to define dynamic similarity than to say what it applied to. Perhaps there is room for both here, but a definition is necessary.

L31 - What does "DFA" stand for? Define or leave it for the main text.

L52 - Please offer your concise definition of dynamic similarity before getting into how it may be quantified.

L55 - "dimensionless ratio" of what?

L86 - Unclear meaning of "distinctive peaks" – "peaks" of what?

L97 - If you're going to use "DFA", then offer it here with the full title.

L98 - "Optima" of what metric(s)?

L122 - What is meant by "relative speed"? The speed should not be normalized by anything for a calculation of Re. Also, given that the focus here is on the hydrodynamics of feeding, I would think that the gape diameter would make more sense for a characteristic length. I'm not going to insist that the authors change this, but they need at least to offer a justification for their selection.

L130 - TTPG should not be italicized.

Please unitalicize all units.

L133 - Please justify the use of different characteristic length or, better yet, use the same for both calculations.

L149 - What test demonstrated the normal distributions? Were there enough repeated measures to demonstrate a normal distribution? Alternatively, was it assumed that this transformation is more likely to yield a normal distribution? Please explain.

L153 - Please explain what a "class" is in this context (the guilds?).

L218 - I do not think it is accurate to state that high Re and WN value are "dominated" by piscivores. The piscivores look pretty evenly distributed among their range, which is similar to omnivores.

L239 - I do not understand the basis for “estimated optima” here. If all of the statistics cited in the prior paragraph had to do with estimating an optimum, then I cannot find where that is explained. I cannot find clarity in the Methods either. A lay reader will certainly be more confused than me, so a better explanation is required. As a consequence of my confusion, on this point, I do not understand what Fig. 3 is showing and the legend does't explain either.

L256-268 - Please explain the significance of the within-family patterns. This section is describing Fig. 4, but I am not sure how all of this relates to the major aims of the study.

L276 - Alternative explanation: zooplanktivores are small fish, which means that they have smaller Re and WN values. The omnivores and fish eaters overlap quite a bit, probably b/c they are similar in size. If the authors have performed an analysis that rules out the possibility that they are merely demonstrating the effects of body size, then they should explain. Perhaps a comparison of species that are similar in size, but achieve different WN and Re values thru behavior would help.

L323 - Aren't copepods “small invertebrates”?

It's always appreciated by reviewers when you can provide the legend on the same page as the figure/

Fig 2 legend - Please offer descriptive text for those who are new to a “Phylomorphospace”. e.g., what do the lines between nodes represent? Do you mean “upper left” inset?

Fig 4 legend - Please explain what each panel shows (why are there 5 panels?). Explain what the lines between the species means. Explain why there are fewer species titles than circles.

Fig. 4 -Many of the graphics are rather jumbled and it's difficult to see the details of the graphics. Are the species names necessary?

Review form: Reviewer 3

Recommendation

Accept with minor revision (please list in comments)

Scientific importance: Is the manuscript an original and important contribution to its field?

Excellent

General interest: Is the paper of sufficient general interest?

Acceptable

Quality of the paper: Is the overall quality of the paper suitable?

Good

Is the length of the paper justified?

Yes

Should the paper be seen by a specialist statistical reviewer?

No

Do you have any concerns about statistical analyses in this paper? If so, please specify them explicitly in your report.

No

It is a condition of publication that authors make their supporting data, code and materials available - either as supplementary material or hosted in an external repository. Please rate, if applicable, the supporting data on the following criteria.

Is it accessible?

Yes

Is it clear?

Yes

Is it adequate?

Yes

Do you have any ethical concerns with this paper?

No

Comments to the Author

This manuscript represents a novel way of differentiating trophic guilds, one that is based on real world performance differences. The finding that trophic guilds of these fishes evolve toward hydrodynamically influenced optima makes this paper unique in its treatment of suction feeding.

That said, I would love to see the authors comment on how these methods could potentially be used for other systems. I would like to see them take this beyond a "suction feeding in fishes" paper. They tease at this at the start of the discussion (lines 275-278) when they discuss combining engineering principles with comparative phylogenetic methods.....and again in lines 331-333 as they report their results as potentially showing the interactions between solid structures. I think that the authors need to more clearly articulate how the use of their methods could be used to analyze other systems. I think that without this broader treatment this manuscript will only be of interest to ichthyologists.

Introduction- Aren't Reynolds numbers also involved in the suction feeding event? Thus, they would be involved in both ram and suction events. Does the Womersley number adequately deal with this issue?

Does using only a single-pulse event potentially mischaracterize fishes that may use more than a single-pulse event. I wonder whether fishes trying to dislodge a prey item clinging to the substrate may use multi-pulse events.

Line 72 needs a [before Westneat

Line 85- hypothesis is somewhat redundant given that the sentence starts with We posit

Line 106 need a period at the end of this sentence

Line 286 and 290 should be cross-sectional area

Line 296 "general reduction of facial features should be explained more clearly. Again, it would be good to make this manuscript more reader friendly to an uninitiated audience.

Line 328 remove (in front of Bellwood

Decision letter (RSPB-2021-1383.R0)

13-Jul-2021

Dear Dr Olsson:

I am writing to inform you that your manuscript RSPB-2021-1383 entitled "'Trophic guilds of suction-feeding fish are distinguished by their characteristic hydrodynamics of swimming and feeding'" has, in its current form, been rejected for publication in Proceedings B.

This action has been taken on the advice of referees, who have recommended that substantial revisions are necessary. With this in mind we would be happy to consider a resubmission, provided the comments of the referees are fully addressed. However please note that this is not a provisional acceptance.

Sincerely,
Dr John Hutchinson, Editor
mailto: proceedingsb@royalsociety.org

Associate Editor
Board Member: 1
Comments to Author:
Associate Editor: Doug Altshuler

Olsson et al. have performed a comparative analysis of the hydrodynamics of swimming and feeding, which leads to interesting differentiation among trophic guilds. All three referees (and I) are enthusiastic about the approach and the results. The methods could be applied to other groups of animals so the potential for broad impact is strong. The main results will also be of significant interest to fish biologists. The referees also raised several substantial criticisms that prevents acceptance of the manuscript in its current form. Two of the referees point out that the results could be explained by 1) body size as opposed to 2) hydrodynamics. It would be important to see hypotheses 1 and 2 directly competed in a revised submission. It is also necessary to include primary references for diet category. Lastly, all three referees identified language that requires clarification and better definition. It would be valuable to see if the authors can revise their manuscript according these very helpful and well informed reviews.

Reviewer(s)' Comments to Author:

Referee: 1

Comments to the Author(s)

This is a terrifically interesting manuscript that introduces the use of a pair of dimensionless ratios to describe the mechanical environment of fish feeding strikes, and then uses these to explore the power of the ratios to describe three trophic groups. The result is a strong pattern of separation of trophic group in the space of the two ratios. The paper has great potential because of its novel use of these two ratios to gain insight into fish feeding diversity. However there are two major issues that have left me less than enthusiastic about the present version of this paper. First, I was confused by some of the categorization of species by diet category. When I looked up several species that I questioned I found data that supported a different category than used in this manuscript. Citations of primary literature are needed for every species – the studies that documented what the fish eat not just books and reviews. Second, my impression is that body size is positively correlated with each of the dimensionless ratios and it is well known that piscivores tend to be large compared to invertivores. So it seems to me that body size may be a trivial explanation for the empirical results presented here. This is the most important issue that needs to be addressed.

L31. The word 'acanthomorph' should not be capitalized. Capitalize it if you use the formal group name – Acanthomorpha, but not for the informal name acanthomorph.

L48. I don't understand this last phrase of the sentence – that thrust from the fins 'determines feeding rate'? It seems as though you are saying that the amount of thrust that the fins generate limits the feeding rate, but that does not seem very likely. This either needs to be clarified or simplified.

L53. You place the term 'similitude' in parentheses here, as another way of saying dynamic similarity, but then this word never appears again in the paper. Perhaps you could just drop it here?

L53-55. The grammar of this sentence is awkward. It reads that you are saying that dynamic similarity can be applied as a dimensionless ratio? Surely that's not what you intend?

L56. It is not clear what dimensionless numbers you are referring to here. This section needs to be rewritten.

L67. Has suction feeding been difficult to 'unravel'? Or do you mean that it has resisted attempts to understand the role of dynamic similarity in shaping patterns of its diversity?

L95. Omnivore refer to an animal that eats both plant and animal prey. Is this really what you mean? This is not a 'feeding guild' on the same scale as 'piscivore' or 'zooplanktivore'.

L102-. I am concerned about the placement of species into diet categories. There are several issues. First, what do the authors mean by 'omnivore'. This term refers to an animal that eats animals and plants, but there are many 'omnivores' in this study that do not eat plants. Second, exactly how was diet category determined? You cannot just use what different authors say because they may each have a different definition of a piscivore. I looked up half a dozen species from the study that I questioned and I could not find data that supported their categorization in this study (*Thorichthys meeki*, *Trichromis salvini*, *Lepomis humilis*, *Lepomis gulosus*, *Lepomis symmetricus*, *Cichlasoma trimaculatum*, *Plectranthias inermis*). So exactly how were fish categorized for this study? What criteria makes a species piscivorous? Zooplanktivorous? And what do you mean 'omnivore'? If that is simply a species that is neither a planktivore or a piscivore then it is such a huge, ecological diverse category that I find it very hard to justify.

L104. The Hulsey et al. 2010 paper used for cichlid data does not have any kinematic timing data in it so it is unclear what is meant here that the data used in the present study was taken from that study?

L142. Is it not important to have length data from the individual fish used for timing data etc? Using length data from fishbase is likely to only be accurate to within an order of magnitude. Lab work is normally done on small specimens and adults in the wild are almost always larger than the typical lab experimental fish. This needs to be justified much more fully.

L150. ‘...to one measure for each species...’ This is an average – not a ‘measure’.

L214. Figures 3 & 4. I am concerned that body size is an awkward complication in both the diet data and the dimensionless ratios. On average we would expect larger fish to have higher Reynolds numbers. What is the relationship between Re and body size of the specimens used in the videos?

L214. In a similar way, it appears that the Womersley number calculated here would be positively correlated with body size of the specimens filmed?

L214. And finally, piscivores tend to be larger than invertivores. So, is body size a trivial explanation for the distinction that Womersley number and Re produce between invertivores and piscivores?

Figure 3. There are some things that I find confusing when I compare figure 3 and figure 4. The point for *Lepomis punctatus* in fig 4 (a species mean) is positioned outside the range of all observations on omnivores in figure 3. How can that be. The same is true for the piscivorous cichlid *Petenia splendida* and many other piscivorous cichlids. Species means that fall outside the range of individual observations for that trophic group?? I assume that this has something to do with this being a phylomorphospace but honestly it just does not make sense.

Figure 4. This is a very helpful figure but about half of the species labels are vague about which data point they are meant to label, and many are positioned awkwardly on top of the outlines of the peaks from the OU analysis. This absolutely needs to be cleaned up. Ideally I also recommend that you label more species. I also suggest that the phylogeny in this figure adds very little, so it might help to remove it and possible use lines to connect some points to the names.

Referee: 2

Comments to the Author(s)

Ollson et al have performed a meta-analysis of fish feeding and swimming in the interest of understanding how species from different guilds have evolved with respect to the hydrodynamics of feeding and swimming. I am excited by the premise of this study and very-much like the idea of exploring patterns of diversity with respect to non-dimensional constants from the physical sciences. I think this subject would interest the readers of Proc B and the novel perspective on understanding fish diversity has the potential for a large impact on an area that interests many organismal biologists.

However, I have the following major concerns about this paper.

1. A major component of the analysis of this study generates “trait optima” that are not explained well enough for readers like me to understand their basis. This is arguably the most important feature of the study and it is mysterious to me how it came about and what it means.

2. It seems possible that the results are merely the consequence of the size differences among species. Re and Womersley number (WN) depend on linear measurements of size. The body

length (which factors into Re) is probably well-correlated with the gape diameter (which goes into WN). In contrast, the authors discuss Re and WN as if they are independent metrics of swimming and feeding. Since the prey for a species also depends on size, then it would appear possible that size is what predicts guild, Re , and WN , above all other factors. If I am wrong about this, then the paper needs to articulate why that interpretation (which may readers will share) is incorrect.

3. As itemized below, there are many areas of the writing in need of clear definitions and explanations of the justification and/or significance.

SPECIFIC COMMENTS

L27 - I'm not sure what "transition in trophic guilds" means. If "guild" refers to a group of species that exploit a particular niche, then how does a species "transition in" one of those? Please expand or rephrase.

L28 - It would be more valuable to define dynamic similarity than to say what it applied to. Perhaps there is room for both here, but a definition is necessary.

L31 - What does "DFA" stand for? Define or leave it for the main text.

L52 - Please offer your concise definition of dynamic similarity before getting into how it may be quantified.

L55 - "dimensionless ratio" of what?

L86 - Unclear meaning of "distinctive peaks" – "peaks" of what?

L97 - If you're going to use "DFA", then offer it here with the full title.

L98 - "Optima" of what metric(s)?

L122 - What is meant by "relative speed"? The speed should not be normalized by anything for a calculation of Re . Also, given that the focus here is on the hydrodynamics of feeding, I would think that the gape diameter would make more sense for a characteristic length. I'm not going to insist that the authors change this, but they need at least to offer a justification for their selection.

L130 - TTPG should not be italicized.

Please unitalicize all units.

L133 - Please justify the use of different characteristic length or, better yet, use the same for both calculations.

L149 - What test demonstrated the normal distributions? Were there enough repeated measures to demonstrate a normal distribution? Alternatively, was it assumed that this transformation is more likely to yield a normal distribution? Please explain.

L153 - Please explain what a "class" is in this context (the guilds?).

L218 - I do not think it is accurate to state that high Re and WN value are "dominated" by piscivores. The piscivores look pretty evenly distributed among their range, which is similar to omnivores.

L239 - I do not understand the basis for "estimated optima" here. If all of the statistics cited in the prior paragraph had to do with estimating an optimum, then I cannot find where that is

explained. I cannot find clarity in the Methods either. A lay reader will certainly be more confused than me, so a better explanation is required. As a consequence of my confusion, on this point, I do not understand what Fig. 3 is showing and the legend doesn't explain either.

L256-268 - Please explain the significance of the within-family patterns. This section is describing Fig. 4, but I am not sure how all of this relates to the major aims of the study.

L276 - Alternative explanation: zooplanktivores are small fish, which means that they have smaller Re and WN values. The omnivores and fish eaters overlap quite a bit, probably b/c they are similar in size. If the authors have performed an analysis that rules out the possibility that they are merely demonstrating the effects of body size, then they should explain. Perhaps a comparison of species that are similar in size, but achieve different WN and Re values thru behavior would help.

L323 - Aren't copepods "small invertebrates"?

It's always appreciated by reviewers when you can provide the legend on the same page as the figure/

Fig 2 legend - Please offer descriptive text for those who are new to a "Phylomorphospace". e.g., what do the lines between nodes represent? Do you mean "upper left" inset?

Fig 4 legend - Please explain what each panel shows (why are there 5 panels?). Explain what the lines between the species means. Explain why there are fewer species titles than circles.

Fig. 4 -Many of the graphics are rather jumbled and it's difficult to see the details of the graphics. Are the species names necessary?

Referee: 3

Comments to the Author(s)

This manuscript represents a novel way of differentiating trophic guilds, one that is based on real world performance differences. The finding that trophic guilds of these fishes evolve toward hydrodynamically influenced optima makes this paper unique in its treatment of suction feeding.

That said, I would love to see the authors comment on how these methods could potentially be used for other systems. I would like to see them take this beyond a "suction feeding in fishes" paper. They tease at this at the start of the discussion (lines 275-278) when they discuss combining engineering principles with comparative phylogenetic methods.....and again in lines 331-333 as they report their results as potentially showing the interactions between solid structures. I think that the authors need to more clearly articulate how the use of their methods could be used to analyze other systems. I think that without this broader treatment this manuscript will only be of interest to ichthyologists.

Introduction- Aren't Reynolds numbers also involved in the suction feeding event? Thus, they would be involved in both ram and suction events. Does the Womersley number adequately deal with this issue?

Does using only a single-pulse event potentially mischaracterize fishes that may use more than a single-pulse event. I wonder whether fishes trying to dislodge a prey item clinging to the substrate may use multi-pulse events.

Line 72 needs a [before Westneat

Line 85- hypothesis is somewhat redundant given that the sentence starts with We posit

Line 106 need a period at the end of this sentence

Line 286 and 290 should be cross-sectional area

Line 296 "general reduction of facial features should be explained more clearly. Again, it would be good to make this manuscript more reader friendly to an uninitiated audience.

Line 328 remove (in front of Bellwood

Author's Response to Decision Letter for (RSPB-2021-1383.R0)

See Appendix A.

RSPB-2021-1968.R0

Review form: Reviewer 1

Recommendation

Major revision is needed (please make suggestions in comments)

Scientific importance: Is the manuscript an original and important contribution to its field?

Excellent

General interest: Is the paper of sufficient general interest?

Acceptable

Quality of the paper: Is the overall quality of the paper suitable?

Marginal

Is the length of the paper justified?

Yes

Should the paper be seen by a specialist statistical reviewer?

No

Do you have any concerns about statistical analyses in this paper? If so, please specify them explicitly in your report.

No

It is a condition of publication that authors make their supporting data, code and materials available - either as supplementary material or hosted in an external repository. Please rate, if applicable, the supporting data on the following criteria.

Is it accessible?

Yes

Is it clear?

Yes

Is it adequate?

Yes

Do you have any ethical concerns with this paper?

No

Comments to the Author

This is a revision of a ms I reviewed earlier. The revision is definitely improved but the main issues remain. Body size is correlated with all three factors being investigated, Reynolds number, Womersley number and diet. Although the results are tantalizing, it is not entirely clear that diet shapes the hydrodynamic regime beyond their shared association with scale. There are also some other issues that I want to raise with the new version. These include concerns about the fact that the size of films fish often does not match adult sizes of those fish for which diet came from, and I found some confusing issues in several figures that would need to be addressed.

The core idea in the paper is that the hydrodynamic regime of suction feeding is characteristic of very general feeding guilds – zooplanktivores, piscivores and generalist invertivores. Two non-dimensional numbers are calculated based on videos of 71 species of fish feeding in the lab and these values are evaluated with respect to diet habits – Reynolds number and Womersley number. A discriminant analysis only correctly classifies species into diet category about 50% of the time. Continuous trait model fitting supports the presence of distinct ‘adaptive peaks’ within each dimensionless number for each of the three diet categories. The implication is that feeding on fish prey results in adaptations that result in higher Re and higher Womersley number, while adaptation to lower values of both characterizes planktivores.

The biggest concern offered in my earlier review (and also put forth by the other reviewer) was that Reynolds number, Womersley number and diet can all be expected to be highly correlated with body size. The authors responded to this by showing that their OU model fitting provides a better fit to the two dimensionless numbers than to body size or mouth gape. That is partly satisfying, but I am not convinced by this response. I think the better way to deal with it would be to actually remove the effects of body size from both numbers and to then show that diet regime affects these residuals. The authors did not address the point that diet is associated with size - zooplankton are small and fish prey are large so zooplanktivores tend to be small and piscivores tend to be large. There is a very large literature that shows that both within species (during ontogeny) and across species, piscivory is associated with larger body size (and larger gape). In order to demonstrate adaptation to prey-specific hydrologic regimes, I think you need to remove the effect of body size from the variables that go into these dimensionless numbers to show that size-corrected time to peak gape, ram speed, and gape size combine in a way that shows these different hydrologic regimes independent of body size.

A new concern is that in this manuscript there is a complex pattern of mismatch between the size of fish upon which determinations of diet were made and the size of specimens used in the video trials. The cichlids are all rather small individuals ranging from 65-95 mm but the centrarchids ranged from 45 to 300 mm and the serranids from 35 to 250. Some of the individuals filmed were radically different from typical adult sizes of the species (which would have been the size fish that diet was determined from). For example the serranid *Variola louti* filmed was 55 mm but a typical adult of this species would be at least 10 times that length. It is not even clear if this species is piscivorous when it is 55 mm – few fish species are piscivores at such a small body size. If the two dimensionless numbers are both strongly affected by body size, then clearly observations on this species should not be taken as representative of adults that are piscivores.

I had not appreciated previously that many of the species values are based on a single sequence for a single species. That is true for all the cichlid species and is worrisome given that kinematics varies from strike to strike and motivation affects ram speed and time to peak gape.

L47. This sentence would need to be rewritten. “There, in which the behaviors...” does not make sense.

L109. acanthomorph, not acantomorph

L216-230. I find that I am wondering if you have sufficient power with this data set to allow you to use the full multi-peak OU model that allows both sigma and alpha to vary? The results of simulation studies that demonstrate sufficient power should be included in the SI. I confess that it does not really matter as I do think there is sufficient power to distinguish BM, single peak OU and multi-peak OU and that is really what matters. But given all the sources of error in this study that are not being accounted for I don't think you should be taking the different estimates of sigma and alpha very seriously.

L280. These descriptions of the locations of the adaptive peaks in the OU modeling are not consistent with figure 3, which shows the peak for generalists has the highest Reynolds number and highest Womersley number, not the piscivores as stated here. Indeed the relative position of the three peaks is surprisingly inconsistent in different parts of the paper. For example, I just get completely confused about what is going on when I compare Figure 3, Figure SI 3, and Figure SI 4. I have to say - something seems off here. Maybe there is a mistake in the color coding in some of the figures?

Figure SI 4. Looking at the plots for cichlids and serranids I am struck by how poor the match of species means are to the appropriate adaptive peaks. In serranids none of the piscivores fall within the 75% confidence lines of the piscivore peak but three species fall within the 75% limits of the generalist peak. The cichlids also seem to be poor matches with the location of the peaks. It really makes it hard to take seriously the locations of the peak when they do not seem to match the empirical data very well.

Figure SI 1. I am having a hard time seeing how this cluster analysis supports the diet categories. Unless I misunderstand the figure, it shows that piscivores fall out in all four major clusters and that planktivores fall out in three of the four clusters. This would appear to be a very weak match between the three named diet categories and the stomach content data used to make this figure.

Review form: Reviewer 2

Recommendation

Accept as is

Scientific importance: Is the manuscript an original and important contribution to its field?

Good

General interest: Is the paper of sufficient general interest?

Good

Quality of the paper: Is the overall quality of the paper suitable?

Good

Is the length of the paper justified?

Yes

Should the paper be seen by a specialist statistical reviewer?

No

Do you have any concerns about statistical analyses in this paper? If so, please specify them explicitly in your report.

No

It is a condition of publication that authors make their supporting data, code and materials available - either as supplementary material or hosted in an external repository. Please rate, if applicable, the supporting data on the following criteria.

Is it accessible?

Yes

Is it clear?

Yes

Is it adequate?

Yes

Do you have any ethical concerns with this paper?

No

Comments to the Author

I am satisfied that the authors have addressed my concerns.

Decision letter (RSPB-2021-1968.R0)

04-Oct-2021

Dear Dr Olsson:

Your manuscript has now been peer reviewed and the reviews have been assessed by an Associate Editor. The reviewers' comments (not including confidential comments to the Editor) and the comments from the Associate Editor are included at the end of this email for your reference. As you will see, the reviewers and the Editors have raised some concerns with your manuscript and we would like to invite you to revise your manuscript to address them.

Research ethics:

Use of animals and field studies:

It is a condition of publication that you make available the data and research materials supporting the results in the article (<https://royalsociety.org/journals/authors/author-guidelines/#data>). Datasets should be deposited in an appropriate publicly available repository and details of the associated accession number, link or DOI to the datasets must be included in the Data Accessibility section of the article (<https://royalsociety.org/journals/ethics-policies/data-sharing-mining/>). Reference(s) to datasets should also be included in the reference list of the article with DOIs (where available).

Please submit a copy of your revised paper within three weeks. If we do not hear from you within this time your manuscript will be rejected. If you are unable to meet this deadline please let us know as soon as possible, as we may be able to grant a short extension.

Best wishes,
Dr John Hutchinson, Editor
mailto: proceedingsb@royalsociety.org

Associate Editor Board Member
Comments to Author:
Associate Editor: Doug Altshuler

The resubmitted manuscript on the hydrodynamics of suction-feeding fish has now been seen by the two referees that raised the most substantial concerns in the first round of review. One referee is satisfied, while the other makes the case that there are two outstanding issues. The first concern is that the effect of body size has not been accounted for, as was originally requested by two of the three referees. The second major concern is the inclusion of new information raises concerns about whether the size of the fish that were filmed are actually being reported correctly. Their view is that the sizes of some fish are outside of the normal range for the different taxa. Both concerns are significant and will need to be addressed before we could consider publication. However, the referee provides good advice for handling body mass confounds, and also for clarifying the other points of confusion.

Reviewer(s)' Comments to Author:

Referee: 1

Comments to the Author(s).

This is a revision of a ms I reviewed earlier. The revision is definitely improved but the main issues remain. Body size is correlated with all three factors being investigated, Reynolds number, Womersley number and diet. Although the results are tantalizing, it is not entirely clear that diet shapes the hydrodynamic regime beyond their shared association with scale. There are also some other issues that I want to raise with the new version. These include concerns about the fact that the size of films fish often does not match adult sizes of those fish for which diet came from, and I found some confusing issues in several figures that would need to be addressed.

The core idea in the paper is that the hydrodynamic regime of suction feeding is characteristic of very general feeding guilds – zooplanktivores, piscivores and generalist invertivores. Two non-dimensional numbers are calculated based on videos of 71 species of fish feeding in the lab and these values are evaluated with respect to diet habits – Reynolds number and Womersley number. A discriminant analysis only correctly classifies species into diet category about 50% of the time. Continuous trait model fitting supports the presence of distinct ‘adaptive peaks’ within each dimensionless number for each of the three diet categories. The implication is that feeding on fish prey results in adaptations that result in higher Re and higher Womersley number, while adaptation to lower values of both characterizes planktivores.

The biggest concern offered in my earlier review (and also put forth by the other reviewer) was that Reynolds number, Womersley number and diet can all be expected to be highly correlated with body size. The authors responded to this by showing that their OU model fitting provides a better fit to the two dimensionless numbers than to body size or mouth gape. That is partly satisfying, but I am not convinced by this response. I think the better way to deal with it would be to actually remove the effects of body size from both numbers and to then show that diet regime affects these residuals. The authors did not address the point that diet is associated with size - zooplankton are small and fish prey are large so zooplanktivores tend to be small and piscivores tend to be large. There is a very large literature that shows that both within species (during ontogeny) and across species, piscivory is associated with larger body size (and larger gape). In order to demonstrate adaptation to prey-specific hydrologic regimes, I think you need to remove the effect of body size from the variables that go into these dimensionless numbers to

show that size-corrected time to peak gape, ram speed, and gape size combine in a way that shows these different hydrologic regimes independent of body size.

A new concern is that in this manuscript there is a complex pattern of mismatch between the size of fish upon which determinations of diet were made and the size of specimens used in the video trials. The cichlids are all rather small individuals ranging from 65-95 mm but the centrarchids ranged from 45 to 300 mm and the serranids from 35 to 250. Some of the individuals filmed were radically different from typical adult sizes of the species (which would have been the size fish that diet was determined from). For example the serranid *Variola louti* filmed was 55 mm but a typical adult of this species would be at least 10 times that length. It is not even clear if this species is piscivorous when it is 55 mm - few fish species are piscivores at such a small body size. If the two dimensionless numbers are both strongly affected by body size, then clearly observations on this species should not be taken as representative of adults that are piscivores.

I had not appreciated previously that many of the species values are based on a single sequence for a single species. That is true for all the cichlid species and is worrisome given that kinematics varies from strike to strike and motivation affects ram speed and time to peak gape.

L47. This sentence would need to be rewritten. "There, in which the behaviors..." does not make sense.

L109. acanthomorph, not acantomorph

L216-230. I find that I am wondering if you have sufficient power with this data set to allow you to use the full multi-peak OU model that allows both sigma and alpha to vary? The results of simulation studies that demonstrate sufficient power should be included in the SI. I confess that it does not really matter as I do think there is sufficient power to distinguish BM, single peak OU and multi-peak OU and that is really what matters. But given all the sources of error in this study that are not being accounted for I don't think you should be taking the different estimates of sigma and alpha very seriously.

L280. These descriptions of the locations of the adaptive peaks in the OU modeling are not consistent with figure 3, which shows the peak for generalists has the highest Reynolds number and highest Womersley number, not the piscivores as stated here. Indeed the relative position of the three peaks is surprisingly inconsistent in different parts of the paper. For example, I just get completely confused about what is going on when I compare Figure 3, Figure SI 3, and Figure SI 4. I have to say - something seems off here. Maybe there is a mistake in the color coding in some of the figures?

Figure SI 4. Looking at the plots for cichlids and serranids I am struck by how poor the match of species means are to the appropriate adaptive peaks. In serranids none of the piscivores fall within the 75% confidence lines of the piscivore peak but three species fall within the 75% limits of the generalist peak. The cichlids also seem to be poor matches with the location of the peaks. It really makes it hard to take seriously the locations of the peak when they do not seem to match the empirical data very well.

Figure SI 1. I am having a hard time seeing how this cluster analysis supports the diet categories. Unless I misunderstand the figure, it shows that piscivores fall out in all four major clusters and that planktivores fall out in three of the four clusters. This would appear to be a very weak match between the three named diet categories and the stomach content data used to make this figure.

Referee: 2

Comments to the Author(s).

I am satisfied that the authors have addressed my concerns.

Author's Response to Decision Letter for (RSPB-2021-1968.R0)

See Appendix B.

RSPB-2021-1968.R1

Review form: Reviewer 1

Recommendation

Major revision is needed (please make suggestions in comments)

Scientific importance: Is the manuscript an original and important contribution to its field?

Excellent

General interest: Is the paper of sufficient general interest?

Good

Quality of the paper: Is the overall quality of the paper suitable?

Excellent

Is the length of the paper justified?

Yes

Should the paper be seen by a specialist statistical reviewer?

No

Do you have any concerns about statistical analyses in this paper? If so, please specify them explicitly in your report.

No

It is a condition of publication that authors make their supporting data, code and materials available - either as supplementary material or hosted in an external repository. Please rate, if applicable, the supporting data on the following criteria.

Is it accessible?

Yes

Is it clear?

Yes

Is it adequate?

Yes

Do you have any ethical concerns with this paper?

No

Comments to the Author

This manuscript introduces a novel line of thinking about fish feeding biomechanics and uses existing data to test the idea that hydrodynamic regime varies with the feeding niche in suction feeding fishes. I really have to give the authors tremendous credit for developing these ideas because I see how this can be a very influential approach. The empirical tests presented in the manuscript indicate that piscivores are operating at higher Reynolds number and higher

Womersley number than zooplanktivores. The results are driven by characteristic differences in morphology and kinematics between the guilds – piscivores have larger mouths, faster approach speeds, and faster mouth opening when capturing prey.

The authors have responded to and addressed my previous concerns quite convincingly and I have no further concerns about the paper.

While I still think it is problematic to represent a piscivorous species with a 55 mm individual that may or may not be a piscivore in nature, I acknowledge that they are likely to express species-typical behavior even at this size. Also, the broadly overlapping size range of fish used in the study, although not representative of the typical pattern in nature among adults where piscivores are larger than zooplanktivores, has the positive effect in this study of showing that even very small-bodied piscivores can demonstrate the guild-specific hydrodynamic regime. In other words, it helps show that the hydrodynamic distinction between the guilds is not simply a trivial consequence of differences in body size between zooplanktivores and piscivores.

I congratulate the authors on this highly original piece of work and a nice paper.

Decision letter (RSPB-2021-1968.R1)

06-Dec-2021

Dear Dr Olsson

I am pleased to inform you that your manuscript entitled "Trophic guilds of suction-feeding fish are distinguished by their characteristic hydrodynamics of swimming and feeding" has been accepted for publication in *Proceedings B*. Congratulations!!

Data Accessibility section

Open Access

Paper charges

Sincerely,

Dr John Hutchinson

Associate Editor:

Comments to Author:

Associate Editor: Doug Altshuler

The manuscript by Olsson et al. has now been seen by three referees over three rounds of review. The consensus from the beginning has been that this is a fascinating study of the hydrodynamics underlying fish feeding and swimming. The explanatory scope of the work is significant, potentially much beyond specific data in the study. The referees had a number of constructive suggestions and the authors have responded extremely well in each round. I congratulate them on an excellent study. I predict this manuscript will be widely read in the fields of fish biology and comparative biomechanics.

Appendix A

Response to reviewers

We thank our reviewers for their work, both for their encouraging remarks and for identifying aspects of the manuscript that ought to be developed to improve clarity, readability, and analytical thoroughness. On considering the points raised by our reviewers, as well as by the Associate Editor, we noticed that certain aspects of the study were raised multiple times, expressing similar concerns. These include the definition of dynamic similarity and the use of the corresponding dimensionless numbers; the caveat that size (rather than Reynolds and Womersley numbers) can better explain the trends; the assignment of feeding guild and diet source data; and the meaning of optima (adaptive peaks) in the context used here (i.e., phylogenetic comparative methods). We would like to begin our response letter by addressing these, before detailing our point-by-point response to the comments by the Editor and the reviewers (see below). We upload a track-change version of the originally submitted manuscript as ESM and a clean version as the main document.

1) Questions were raised with respect to the definition of dynamic similarity, and the meaning and application of the dimensionless numbers.

We have expanded on the definition of dynamic similarity and the use of dimensionless numbers in the Introduction (**L55-72**). Dynamic similarity corresponds to two or more cases of flow (two hydrodynamic regimes) if the distribution of forces (i.e.: inertial, viscous, unsteady, etc.) they experience is similar, i.e., if the corresponding forces are proportional, relate in magnitude and scale by a constant factor. Hence, dynamic similarity is employed in the analysis of situations where mass is in motion (momentum). Mathematically, the distribution of forces can be expressed as a ratio, i.e., the dimensionless number. The nature of the flow determines the appropriate utilization of specific dimensionless number(s): for example, Reynolds number applies to situations of steady flow, Strouhal number to undulatory motion, and Womersley number to pulsating flow. Dynamic similarity is thus assessed using the dimensionless numbers relevant to the hydrodynamic forces that dominate the studied system, and is valid if these numbers agree.

2) Reviewers and the Editor expressed concern that the observed pattern is due to body size (i.e., standard length), and the correlation between body size and the Reynolds number

We agree that body size is a key parameter in organismal biology, known to affect numerous physiological and ecological processes. Moreover, the dimensionless numbers used in this paper, Re and

Wom, are directly calculated using the characteristic lengths (standard length for Re and peak gape for Wom) as one of their input variables (eqs 1-2 in the manuscript). This necessarily results in a relationship between size and the hydrodynamic regime, and inevitably to size explaining at least some of the variation in the evolutionary models we fitted. A general point to make is that while body size indeed affects numerous other processes, this ubiquity somewhat masks a specific, functional link between body size and feeding ecology. In contrast, a hydrodynamic regime has a precise definition. Our approach was to ask if size alone is an equally good explanatory variable as the hydrodynamic regime; or whether the hydrodynamic regime conveys more information about the evolution of feeding guilds (**L231-243**). We formally assessed whether size alone could explain our results by comparing the AICc value for an evolutionary model in which the explanatory variable was size (SL or gape size) to that of a model in which the explanatory variable was the dimensionless number (Re or Wom). If size alone encapsulates all the hydrodynamic phenomena, the difference in AICc should be zero. A lower AICc value for the model fitted using the dimensionless number would indicate that the hydrodynamic regime carries additional information beyond that encompassed by size alone. Indeed, we found that the median AICc was lower for models in which the dimensionless numbers were the explanatory variables compared to models with lengths (**L297-308, ESM 4.1, Table SI 2, Fig. SI 2**). Thus, we conclude that hydrodynamic traits have greater explanatory power than lengths in the context of the evolution of feeding guilds.

3) Reviewers raised questions regarding the method for assigning feeding guild (requesting more precise use of sources), along with the term 'omnivore'.

We agree with the criticism of 'omnivore' and have replaced it with 'generalist' throughout the manuscript. We have revisited the diet data on the species in our data set to locate the original sources (**ESM 2, Table SI 1**). We prioritized sources that reported stomach content data. However, for a subset of 19 species such sources could not be found. For these cases we located at least two sources that agreed on the same feeding guild, determined based on behavioral studies, personal observation, or similarity to closely related species. As part of the process one species (*T. meeki*) was reassigned from zooplanktivore to molluskivore (the original source clearly identified zooplanktivory, but a subsequently consulted source indicated that the transition to a gastropod diet is strongly linked to body size). We verified the assignment of species to feeding guilds using a dissimilarity analysis of the stomach content data (**ESM Fig. SI 1**). This analysis showed good agreement between the feeding guild we originally assigned and the dissimilarity in diet data. This agreement was particularly high for zooplanktivores and

herbivores. Marine generalists tended to cluster closely with marine piscivores while freshwater generalists clustered more closely with freshwater piscivores, presumably reflecting the relative abundance of different prey types in marine and freshwater habitats. To assess if this changed had an impact on our results, we split the generalists into freshwater and marine generalists and performed the analyses on the resultant four guilds. This classification still resulted in highest support for multi-peak models (**L146-153, ESM 4.2, Table SI 3**), and a visual examination of the distribution of peaks revealed that optimal regions for piscivores and zooplanktivores were relatively unchanged while the optimal regions for freshwater and marine generalists overlapped considerably (**ESM Fig. SI 3**).

4) Reviewers found the use of the term 'optima' unclear, along with the associated analyses, as well as the result illustrated in Fig. 3

We have expanded the explanation of the phylogenetic comparative methods and the terminology associated with them (**L172-228**). We clarify that the hypothesis of our study is that different feeding guilds evolve towards different adaptive peaks, i.e., the optimal combination of traits (here, Re and Wom ; **L92-94**). Under this hypothesis, species evolve towards different Re and Wom depending on what kind of prey they target. We further explain that the existence of an adaptive peak, and the possibility of multiple adaptive peaks that are unique to different guilds, can be assessed by fitting phylogenetic models that estimate the likelihood that a set of parameters (the location of trait optima, pull towards optima and the rate of the Brownian motion; **L179-182, L83-201**), are common to all guilds or specific to each guild. We then explain that the distribution of these estimated optima can be used to characterize the existence and location of the adaptive peak, and infer if it is specific to each feeding guild.

Furthermore, we clarified Fig. 3, (**Legend Fig. 3**) which shows only the distribution of the optima. (We also clarified Fig. 4, but moved this result to the ESM and is renamed **Fig. SI 4**, which shows where the extant species are located relative to the estimated optima using a phylomorphospace means of plotting, which we explain is the projection of the phylogenetic tree in trait space; **Legend Fig. SI 4**. We have redrawn **Fig. SI 4** to make it more easily interpretable, though more species labels were found to clutter the overall appearance, and we point to **ESM Table SI 1** for the individual species data). We note that the distribution of extant species in trait space does not map perfectly to the location of the adaptive peaks towards which they evolve, since it depends on the strength of the selective forces that operate on it, as well as on its phylogenetic history (**Legend Fig. 2**).

Below we address each point raised by the Editor and the reviewers.

Associate Editor

Board Member: 1

Comments to Author:

Associate Editor: Doug Altshuler

Olsson et al. have performed a comparative analysis of the hydrodynamics of swimming and feeding, which leads to interesting differentiation among trophic guilds. All three referees (and I) are enthusiastic about the approach and the results. The methods could be applied to other groups of animals so the potential for broad impact is strong. The main results will also be of significant interest to fish biologists. The referees also raised several substantial criticisms that prevents acceptance of the manuscript in its current form. Two of the referees point out that the results could be explained by 1) body size as opposed to 2) hydrodynamics. It would be important to see hypotheses 1 and 2 directly competed in a revised submission. It is also necessary to include primary references for diet category. Lastly, all three referees identified language that requires clarification and better definition. It would be valuable to see if the authors can revise their manuscript according to these very helpful and well informed reviews.

Thank you for this encouraging assessment. We recognize that there is a link between size and Reynolds number (as well as between gape size and Womersley number), and agree that comparing the two approaches ('hydrodynamic' vs 'size') makes an important contribution to the analysis of the data and the strength of the conclusions. As discussed in **point 2 above**, we carried out this comparison, by fitting the seven phylogenetic models to (a) the hydrodynamic traits (Reynolds number and Womersley number) and (b) the size traits (standard length and gape size). For each trait, we calculated the median AICc, log-likelihood and Akaike weight for each of the seven models. This allowed us to identify the best supported model for each hydrodynamic trait and compare it to the same model for the corresponding size trait. This showed that the best model for the hydrodynamic traits outperformed all models for the size traits (**ESM Table SI 2**). For each tree, we also identified the best-performing model for the hydrodynamic trait and compared it to an identically specified model for the size trait. This showed that

the median difference was below 0 (**Fig. SI 2**) emphasizing that a model based on hydrodynamic traits outperformed a model based on size traits.

Reviewer(s)' Comments to Author:

Referee: 1

Comments to the Author(s)

This is a terrifically interesting manuscript that introduces the use of a pair of dimensionless ratios to describe the mechanical environment of fish feeding strikes, and then uses these to explore the power of the ratios to describe three trophic groups. The result is a strong pattern of separation of trophic group in the space of the two ratios. The paper has great potential because of its novel use of these two ratios to gain insight into fish feeding diversity. However there are two major issues that have left me less than enthusiastic about the present version of this paper. First, I was confused by some of the categorization of species by diet category. When I looked up several species that I questioned I found data that supported a different category than used in this manuscript. Citations of primary literature are needed for every species – the studies that documented what the fish eat not just books and reviews. Second, my impression is that body size is positively correlated with each of the dimensionless ratios and it is well known that piscivores tend to be large compared to invertivores. So it seems to me that body size may be a trivial explanation for the empirical results presented here. This is the most important issue that needs to be addressed.

Thank you for the positive response. We agree that the feeding guild assignment requires more details and an additional assessment, see **point 3** above. We revised our sources, searching out primary sources (stomach content analysis) where possible (**L146-152, ESM 2, Table SI 1**). When not possible, we ensured that we had two sources agreeing on the assigned feeding guild (based on personal observations or similarity to related species). We also assessed the assigned guild against recorded food items using a dissimilarity index, concluding good agreement between diet data and assigned guild (**ESM Fig SI 1**). Finally, noting that habitat (marine and freshwater) appeared to influence the type of food items consumed by generalists, we split the generalist guild into freshwater generalists and marine generalists and carried out the analyses using four feeding guilds instead of three. The results obtained

from this analysis were consistent with those obtained when using three guilds (**ESM Table SI 3, Fig. SI 3**).

L31. The word 'acanthomorph' should not be capitalized. Capitalize it if you use the formal group name – Acanthomorpha, but not for the informal name acanthomorph.

This has been done (**L31, also L109**).

L48. I don't understand this last phrase of the sentence – that thrust from the fins 'determines feeding rate'? It seems as though you are saying that the amount of thrust that the fins generate limits the feeding rate, but that does not seem very likely. This either needs to be clarified or simplified.

We have clarified the text as follows: "For example, in herbivorous fish that graze on algae attached to the substrate: the thrust generated by the fins is used to dislodge the food from its holdfast and the speed of reversing away from the substrate determines the amount of algae removed per feeding bout (Perevolotsky et al. 2020)" (**L48-51**).

L53. You place the term 'similitude' in parentheses here, as another way of saying dynamic similarity, but then this word never appears again in the paper. Perhaps you could just drop it here?

We have dropped it (**L55-56**).

L53-55. The grammar of this sentence is awkward. It reads that you are saying that dynamic similarity can be applied as a dimensionless ratio? Surely that's not what you intend?

The paragraph has been largely revised (in response to **point 1 above**). This sentence has been removed (**L55-72**).

L56. It is not clear what dimensionless numbers you are referring to here. This section needs to be rewritten.

As per response above, the paragraph has been rewritten (**L55-72**).

L67. Has suction feeding been difficult to 'unravel'? Or do you mean that it has resisted attempts to understand the role of dynamic similarity in shaping patterns of its diversity?

Thanks for this comment. We have rephrased the sentence: "Suction-feeding in fish is a complex behaviour that has challenged previous efforts to link morphology to diet specialization." (**L73-74**)

L95. Omnivore refer to an animal that eats both plant and animal prey. Is this really what you mean? This is not a 'feeding guild' on the same scale as 'piscivore' or 'zooplanktivore'.

Thanks for this observation. We agree. In the dietary analysis, we identified six guilds: piscivore, zooplanktivore, molluskivore, herbivore, detritivore, and those species consuming a variety of food items. We termed these omnivores, but the better term is generalist, and we have replaced "omnivore" with "generalist" **throughout** the manuscript. In our dissimilarity analysis of feeding guilds, we noted that the diets of marine and freshwater generalists were slightly different, presumably due to availability of different prey types in the two habitats. However, re-running the analysis for four guilds (zooplanktivores, piscivores, marine generalists and freshwater generalists) yielded results consistent with those obtained for three feeding guilds (**ESM Table SI 3**) with the regions containing the adaptive peaks of the two generalist guilds considerably overlapping (**ESM Fig SI 3**).

L102-. I am concerned about the placement of species into diet categories. There are several issues. First, what do the authors mean by 'omnivore'. This term refers to an animal that eats animals and plants, but there are many 'omnivores' in this study that do not eat plants. Second, exactly how was diet category determined? You cannot just use what different authors say because they may each have a different definition of a piscivore. I looked up half a dozen species from the study that I questioned and I could not find data that supported their categorization in this study (*Thorichthys meeki*, *Trichromis salvini*, *Lepomis humilis*, *Lepomis gulosus*, *Lepomis symmetricus*, *Cichlasoma trimaculatum*, *Plectranthias inermis*). So exactly how were fish categorized for this study? What criteria makes a species piscivorous? Zooplanktivorous? And what do you mean 'omnivore'? If that is simply a species that is neither a planktivore or a piscivore then it is such a huge, ecological diverse category that I find it very hard to justify.

Thanks for this comment. We have replaced omnivore with generalist **throughout** the manuscript. As discussed in **point 3 above**, and in reply to the overall remarks from the reviewer on the manuscript, we have revised the diet data and describe in detail how this was done (**L146-152, ESM 2**). All diet sources are given in **ESM Table SI 1**, and the verification of the assigned feeding guild, using a dissimilarity analysis is shown in **ESM Fig SI 1**.

L104. The Hulsey et al. 2010 paper used for cichlid data does not have any kinematic timing data in it so it is unclear what is meant here that the data used in the present study was taken from that study?

We have clarified that we re-analyzed the movies from the Hulseley et al (2010) study to extract all the relevant parameters from the recorded strikes (**L120-121**).

L142. Is it not important to have length data from the individual fish used for timing data etc? Using length data from fishbase is likely to only be accurate to within an order of magnitude. Lab work is normally done on small specimens and adults in the wild are almost always larger than the typical lab experimental fish. This needs to be justified much more fully.

We agree. We ran the analyses (fitting the models, carrying out model averaging to obtain one estimate per tree, and visualized the results) both on the full data set (with lengths obtained from FishBase) as well as from the reduced data set (without FishBase lengths; **L155-158**). This showed that the results obtained from the reduced data set were consistent with those obtained from the full data set (**ESM 4.2, Table SI 3, Fig SI 3**).

L150. ‘...to one measure for each species...’ This is an average – not a ‘measure’.

Agree, we have rephrased the sentence “For each strike we calculated Re and α^2 using Eqs. 1 and 2. To avoid biasing the results towards species for which there were many data points, we reduced the data set to the species average by first calculating the mean Re and α^2 for each individual and then the mean for each species.” (**L158-161**)

L214. Figures 3 & 4. I am concerned that body size is an awkward complication in both the diet data and the dimensionless ratios. On average we would expect larger fish to have higher Reynolds numbers. What is the relationship between Re and body size of the specimens used in the videos?

The relationship between $\log_{10}Re$ and $\log_{10}SL$ is: (phylogenetic generalized linear models) estimate = 0.5, $t = 6.5$, $p < 0.001$, $R^2 = 0.37$. However, as stated in **point 2 above**, there is a relationship between Reynolds number and body size, but a comparative analysis shows that models fitted with hydrodynamic traits (Reynolds and Womersley numbers) have greater explanatory power than models fitted with size traits (standard length and gape size) (**ESM 4.1, Table SI 2, Fig SI 2**). Also please note that we moved Fig. 4 to ESM (now labelled **Fig. SI 4**).

L214. In a similar way, it appears that the Womersley number calculated here would be positively correlated with body size of the specimens filmed?

The relationship between $\log_{10}Wom$ and $\log_{10}SL$ is: (phylogenetic generalized linear models) estimate = 0.47, $t = 8.7$, $p < 0.001$, $R^2 = 0.52$. We refer to **point 2 above**.

L214. And finally, piscivores tend to be larger than invertivores. So, is body size a trivial explanation for the distinction that Womersley number and Re produce between invertivores and piscivores?

Thanks for this question. Here, we refer to the comparative analyses accounted for in **point 2 above** regarding the explanatory power of size traits versus hydrodynamic traits. There is a relationship between size and Reynolds number, but the explanatory power of size traits in evolutionary models is consistently poorer, compared to that of hydrodynamic traits.

Figure 3. There are some things that I find confusing when I compare figure 3 and figure 4. The point for *Lepomis punctatus* in fig 4 (a species mean) is positioned outside the range of all observations on omnivores in figure 3. How can that be. The same is true for the piscivorous cichlid *Petenia splendida* and many other piscivorous cichlids. Species means that fall outside the range of individual observations for that trophic group?? I assume that this has something to do with this being a phylomorphospace but honestly it just does not make sense.

We appreciate your observation. We have rewritten the explanation of the phylogenetic models (**point 4 above, L172-228**). We made it clear that the trait optima (which are plotted in **Fig. 3b**) obtained from the phylogenetic models are estimates of the location of the adaptive peak; i.e., the point (as in, the value of the trait) to which species will evolve given sufficient amount of time. However, species that diverged from a generalist ancestor are unlikely to have had that amount of time and are therefore, plausibly, located away from the peak.

Figure 4. This is a very helpful figure but about half of the species labels are vague about which data point they are meant to label, and many are positioned awkwardly on top of the outlines of the peaks from the OU analysis. This absolutely needs to be cleaned up. Ideally I also recommend that you label more species. I also suggest that the phylogeny in this figure adds very little, so it might help to remove it and possible use lines to connect some points to the names.

We have moved Fig. 4 to ESM and summarize the findings in the main text (**L289-296**). It has been labelled **Fig. SI 4**. We have redrawn the figure to clarify the labelling. We only labelled the species on the edges as labels in the middle would become unintelligible. We judged it interesting to indicate which species occupy the extremes of the trait space. All data can be found in **ESM Table SI 1**. We discuss how

a “messy” look of a phylomorphospace is to be expected, if traits evolve towards different optima in response to a shift in the feeding guild (L294-296).

Referee: 2

Comments to the Author(s)

Ollson et al have performed a meta-analysis of fish feeding and swimming in the interest of understanding how species from different guilds have evolved with respect to the hydrodynamics of feeding and swimming. I am excited by the premise of this study and very-much like the idea of exploring patterns of diversity with respect to non-dimensional constants from the physical sciences. I think this subject would interest the readers of Proc B and the novel perspective on understanding fish diversity has the potential for a large impact on an area that interests many organismal biologists.

We appreciate your insight and grateful for the encouragement.

However, I have the following major concerns about this paper.

1.A major component of the analysis of this study generates “trait optima” that are not explained well enough for readers like me to understand their basis. This is arguably the most important feature of the study and it is mysterious to me how it came about and what it means.

We agree that this part of the manuscript needed a clearer and more detailed description, better linked to the hypotheses and the outcome of the results. We have done so (outlined in **point 3 above, L172-288**). We clarify that feeding strikes can be described hydrodynamically using the Reynolds and Womersley numbers and that the requirements involved in feeding on different types of prey will cause the Reynolds and Womersley numbers to evolve towards different adaptive peaks, i.e., the ideal combinations of traits will differ between the guilds. We further explain that the existence of different peaks can be concluded by comparing the fit (here, the AICc) of phylogenetic models specifying multiple peaks to models specifying single peaks or no peaks. We further explain that the location of such peaks can be inferred from the optima (a parameter estimated when fitting the model) and that with 1000 trees, we obtain 1000 estimates of this location, as plotted in **Fig. 3b**.

2.It seems possible that the results are merely the consequence of the size differences among species. Re and Womersly number (WN) depend on linear measurements of size. The body length

(which factors into Re) is probably well-correlated with the gape diameter (which goes into WN). In contrast, the authors discuss Re and WN as if they are independent metrics of swimming and feeding. Since the prey for a species also depends on size, then it would appear possible that size is what predicts guild, Re , and WN , above all other factors. If I am wrong about this, then the paper needs to articulate why that interpretation (which may readers will share) is incorrect.

As discussed in **point 3 above**, there is a relationship between the Reynolds number and the body size, therefore, at least part of the variation explained by the Reynolds and the Womersley numbers can be explained by the size, too. However, we compared models fitted with hydrodynamic traits (Reynolds and Womersley) to models fitted with size traits (standard length and peak gape) and found that models based on hydrodynamic traits outperformed models based on size traits (**L231-243, L297-308, ESM Table SI 2, Fig. SI 2**). This indicated that the hydrodynamic traits have greater explanatory power than size alone in the context of the evolution of feeding guilds.

3. As itemized below, there are many areas of the writing in need of clear definitions and explanations of the justification and/or significance.

We address these after each point.

SPECIFIC COMMENTS

L27 - I'm not sure what "transition in trophic guilds" means. If "guild" refers to a group of species that exploit a particular niche, then how does a species "transition in" one of those? Please expand or rephrase.

We have rephrased this: "when species evolve to feed on different prey types" (**L27-28**)

L28 - It would be more valuable to define dynamic similarity than to say what it applied to. Perhaps there is room for both here, but a definition is necessary.

The word limit of the Abstract does not allow for the definition here. However, we have rewritten the second paragraph of the introduction to define the concept, explain its use and its relationship to nondimensional numbers (**L55-72**).

L31 - What does "DFA" stand for? Define or leave it for the main text.

DFA stands for discriminant function analysis, but we have removed the acronym in the Abstract (**L31**)

L52 - Please offer your concise definition of dynamic similarity before getting into how it may be quantified.

We define dynamic similarity as “In the case of fluid mechanics, dynamic similarity is said to exist between two flow cases (e.g., the flow generated by small and large organisms in fluids of different viscosities (Cheer and Koehl 1987)) if the forces they experience are parallel, relate in magnitude and scale by a constant factor (Cengel and Cimbala 2006; Fox et al. 2020)” (L56-59)

L55 - “dimensionless ratio” of what?

The formal definition of dimensionless ratio is the ratio of forces exerted by or on a control volume (i.e.: flow domain). The type of forces included depends on the nature of the flow and the interactions within. We explain it in the following manner: “Mathematically, the scaling of the forces is expressed as a ratio, i.e., a dimensionless number; and the nature of the flow determines the appropriate dimensionless number(s) used to assess dynamic similarity. This concept enables a comparison of the hydrodynamics that govern the behaviours of animals of different sizes, speeds, and shapes. For example, swimming in fish is often characterized in terms of the Strouhal number, which provides the ratio of unsteadiness to inertial forces in oscillating flows, and thus links the tail beats to the propulsive efficiency across different sizes and species of fish (Triantafyllou et al. 2000; Taylor et al. 2003).” (L60-67)

L86 - Unclear meaning of “distinctive peaks” — “peaks” of what?

We have rewritten this sentence to clarify that we hypothesize that the “hydrodynamics that characterize suction-feeding evolve towards adaptive peaks that are distinctive for different feeding guilds, i.e., that the combination of traits that optimizes suction feeding differs depending on the targeted prey type.” (L92-94)

L97 - If you’re going to use “DFA”, then offer it here with the full title.

We do not use the acronym (L111)

L98 - “Optima” of what metric(s)?

The optima are the model estimate of the value of the trait (the Reynolds and Womersley numbers) at the adaptive peak; the numbers are dimensionless. This has been rewritten: “We employ phylogenetic comparative methods to test the hypothesis that the Reynolds and Womersley numbers evolve towards

adaptive peaks that are specific to each feeding guild, and quantify the strength of attraction to those peaks. "(L112-115)

L122 - What is meant by "relative speed"? The speed should not be normalized by anything for a calculation of Re . Also, given that the focus here is on the hydrodynamics of feeding, I would think that the gape diameter would make more sense for a characteristic length. I'm not going to insist that the authors change this, but they need at least to offer a justification for their selection.

Agree, we used relative speed as the Reynolds number is (sometimes) defined, i.e., the relative speed between the solid body (here the fish) and the flow of water. As these are laboratory experiments, there was no current in the water and we agree that the word is redundant and confusing here. Removed (L126)

L130 - TTPG should not be italicized. Please unitalicize all units.

We have done so (L130-134).

L133 - Please justify the use of different characteristic length or, better yet, use the same for both calculations.

It is not appropriate to use the same characteristic length for the Reynolds and Womersley numbers, because the parameters should be relevant to the system under investigation. We exemplify this by comparing the calculation of the Reynolds number for the whole body, or for an appendage, in which the appropriate length measure is the whole-body length or the length of the appendage, respectively. We elaborate on our choice in L138-145.

L149 - What test demonstrated the normal distributions? Were there enough repeated measures to demonstrate a normal distribution? Alternatively, was it assumed that this transformation is more likely to yield a normal distribution? Please explain.

The non-transformed Reynolds and Womersley numbers produced significant ($p < 0.05$) Shapiro-Wilks test, while the \log_{10} -transformed values produced $p > 0.1$ using the same test. (L161-163)

L153 - Please explain what a "class" is in this context (the guilds?).

“Class” is the general term used in discriminant functions to describe the category variable. Here, class is the feeding guild. Explained: “Standard linear discriminant analysis assumes that each class (hereafter the three feeding guilds) has a single Gaussian distribution,...” (L165-166)

L218 - I do not think it is accurate to state that high Re and WN value are “dominated” by piscivores. The piscivores look pretty evenly distributed among their range, which is similar to omnivores.

We have rewritten this description for clarification: “The species in our data set are approximately distributed along a slope from low to high values of Re and α^2 . Combinations of low Re and low α^2 (lower left corner in Fig 2) are dominated by zooplanktivores whereas combinations of high Re and high α^2 (upper right corner in Fig 2) are dominated by piscivores” (L247-250)

L239 - I do not understand the basis for “estimated optima” here. If all of the statistics cited in the prior paragraph had to do with estimating an optimum, then I cannot find where that is explained. I cannot find clarity in the Methods either. A lay reader will certainly be more confused than me, so a better explanation is required. As a consequence of my confusion, on this point, I do not understand what Fig. 3 is showing and the legend doesn’t explain either.

We have revised the description of the phylogenetic comparative methods (see **point 2** above), to clarify the meaning of optimum (L172-226). The existence of adaptive peaks can be tested by fitting phylogenetic models with different specifications, and compare the fit of these models using AICc. The optimum is one of the parameters estimated in these models (if the model is specified in such a manner), and corresponds to an estimate of the adaptive peak. Thus, by fitting the models to 1,000 trees we obtain 1,000 model estimates and can use these to locate the region which is likely to contain the adaptive peak.

L256-268 - Please explain the significance of the within-family patterns. This section is describing Fig. 4, but I am not sure how all of this relates to the major aims of the study.

We have moved large parts of this analysis to ESM and reference it briefly in the main text (L289-296). In the full dataset, the distribution of species along the Reynolds – Womersley trait space is affected by the evolutionary history and the selection on the traits in different guilds. In many cases, plotting the trait space for distantly related species results in the evolutionary history obscuring the effect of selection. We assumed that this might have been the case here and sought to provide a more “zoomed-in” view

by plotting each family separately. We think that this helped especially in the cases of the cichlids and centrarchids (**ESM Fig. SI 4**).

L276 - Alternative explanation: zooplanktivores are small fish, which means that they have smaller Re and WN values. The omnivores and fish eaters overlap quite a bit, probably b/c they are similar in size. If the authors have performed an analysis that rules out the possibility that they are merely demonstrating the effects of body size, then they should explain. Perhaps a comparison of species that are similar in size, but achieve different WN and Re values thru behavior would help.

We acknowledge the relationship between body size and the Reynolds number (see **point 2** above). Supplemental analysis of models based on body size versus models based on hydrodynamic traits show that models based on hydrodynamic traits outperformed those based on length, indicating that the hydrodynamic traits carry more information with respect to trait evolution in the context of feeding guilds. (**L231-243, L297-308, ESM Table SI 2, Figure SI 2**)

L323 - Aren't copepods "small invertebrates"?

Copepods are small invertebrates but not all small invertebrates are copepods. This was written to clarify a vague description in a source we used. However, having revised and verified our feeding guild assignment procedure we have rewritten the paragraph and removed this part (**L320-349**).

It's always appreciated by reviewers when you can provide the legend on the same page as the figure/

Agreed. We tried to follow the instructions when submitting but we apologize that we cannot seem to get this correct.

Fig 2 legend - Please offer descriptive text for those who are new to a "Phylomorphospace". e.g., what do the lines between nodes represent? Do you mean "upper left" inset?

Thanks. "Upper left" is right, and we corrected it. Phylomorphospace is the projection of a phylogenetic tree into trait space, and the lines represent the branches of that tree. We have added this information to the **legends of Figs 2 and Fig SI 4**.

Fig 4 legend - Please explain what each panel shows (why are there 5 panels?). Explain what the lines between the species means. Explain why there are fewer species titles than circles.

This Figure has been moved to **ESM 5** and is labelled **Fig SI 4**. The results are referenced more briefly in the main text (**L289-296**). The five panels show the five families (Antennariidae, Cichlidae, Centrarchidae, Pomacentridae, and Serranidae) in our study. The lines are the phylomorphospace, e.g. the projection of the tree into 2D traitspace. Not all points are labelled, so as to reduce the clutter and improve clarity of the plot, while pointing to the ESM to find the values of Re and Wom for each species. **Legend Fig. SI 4** has been rewritten to include this information.

Fig. 4 -Many of the graphics are rather jumbled and it's difficult to see the details of the graphics. Are the species names necessary?

We think that it is informative to show which species occupy the extremes of the trait space. It is also the case that if traits evolve towards different optima following a shift in feeding guild, a phylomorphospace plot will look 'messy' as branches cross over other (Stayton 2020). We explain that this is an aspect of applying the phylogenetic comparative method if traits evolve towards different adaptive peaks, which is the hypothesis we test in our study (**L294-296; Legend Fig 2**). Nevertheless, it is an informative illustration, as it shows the direction in trait space that a species follows, and in this study, shows for example how the branches of zooplanktivorous members of families tend towards the lower left corner (i.e., lower Reynolds and Womersley numbers). We agree that it is better to reduce the amount of 'clutter' in the plots, so we have been more judicious in selecting which points to label by species name. We have also 'zoomed' in on each family to make the distribution of species in trait space clearer. (**Fig. SI 4**)

Referee: 3

Comments to the Author(s)

This manuscript represents a novel way of differentiating trophic guilds, one that is based on real world performance differences. The finding that trophic guilds of these fishes evolve toward hydrodynamically influenced optima makes this paper unique in its treatment of suction feeding.

Thank you for this positive assessment.

That said, I would love to see the authors comment on how these methods could potentially be used for other systems. I would like to see them take this beyond a "suction feeding in fishes" paper. They tease

at this at the start of the discussion (lines 275-278) when they discuss combining engineering principles with comparative phylogenetic methods.....and again in lines 331-333 as they report their results as potentially showing the interactions between solid structures. I think that the authors need to more clearly articulate how the use of their methods could be used to analyze other systems. I think that without this broader treatment this manuscript will only be of interest to ichthyologists.

As our study (as well as the specific hydrodynamics we study and the data that supports it) are on the swimming and suction of fish, we refrained from venturing beyond the subject. Nevertheless, we think that the approach is generalizable, especially to cases in which the link between an ecological function and the forces that characterize performance of that function can be reasonably well hypothesized. This may include the walking on water, the lift/drag ratio and flight performance of wings in birds and bats, and quadrupedal locomotion. We discuss these wider applications at the end of the Discussion (**L378-386**).

Introduction- Aren't Reynolds numbers also involved in the suction feeding event? Thus, they would be involved in both ram and suction events. Does the Womersley number adequately deal with this issue?

The Reynolds number assumes steady flow, but previous work studying the flow generated during suction events indicated steep flow gradients (i.e., accelerations; reviewed in Day et al 2015), which are inconsistent with this assumption and suggest that the Reynolds number may not be entirely appropriate in this context. (**L102-104**)

Does using only a single-pulse event potentially mischaracterize fishes that may use more than a single-pulse event. I wonder whether fishes trying to dislodge a prey item clinging to the substrate may use multi-pulse events.

The steep spatial and temporal gradients are the dominant source for the hydrodynamic force exerted on the prey (Wainwright & Day 2007, Holzman et al 2012), and thus we consider this appropriate in the context of suction feeding (**L104-107**). Other modes of feeding (grazing, biting, excavating) may likely require other considerations.

Line 72 needs a [before Westneat

Done (**L79**)

Line 85- hypothesis is somewhat redundant given that the sentence starts with We posit

Removed; the sentence now reads: “Here we posit that the hydrodynamics that characterize suction-feeding evolve towards adaptive peaks that are distinctive for different feeding guilds, i.e., that the combination of traits that optimizes suction feeding differs depending on the targeted prey type” (L92-94)

Line 106 need a period at the end of this sentence

Done (L123)

Line 286 and 290 should be cross-sectional area

Done (L327, 335)

Line 296 “general reduction of facial features should be explained more clearly. Again, it would be good to make this manuscript more reader friendly to an uninitiated audience.

We have expanded on the results by Schmitz and Wainwright (2011) as well as Friedman et al (2016). This now reads: “In addition to the low Reynolds numbers associated with zooplanktivory in our analysis, previous work on the evolution of zooplanktivory has identified a concomitant reduction in facial features (including jaw length, premaxilla length, distance between the eye, the base of the pectoral fin and the anterior tip of the dentary of the jaw, and the adductor muscle weight; Friedman et al. 2016), and the anterior length and region (demarcated by tip of the premaxilla, anterior orbit margin and the articular-quadrato lower jaw joint; Schmitz and Wainwright 2011), which presumably may contribute to a smaller cross-sectional area.” (L328-335)

Line 328 remove (in front of Bellwood

Done (L367)

Appendix B

Dear Editor,

we appreciate the effort made by you and the reviewers providing us with constructive comments that improved the manuscript quality. We are happy that the majority of the revisions were accepted. As you will see below, in the revised version we have strived to address all the comments of Reviewer #1 and the editor. These are mainly related to the choice of variables used in the phylogenetic models, as well as questions regarding the input data. Our detailed response is provided herein.

Sincerely,

Karin Olsson

Comments to Author:

Editor: Doug Altshuler

The resubmitted manuscript on the hydrodynamics of suction-feeding fish has now been seen by the two referees that raised the most substantial concerns in the first round of review. One referee is satisfied, while the other makes the case that there are two outstanding issues. The first concern is that the effect of body size has not been accounted for, as was originally requested by two of the three referees. The second major concern is the inclusion of new information raises concerns about whether the size of the fish that were filmed are actually being reported correctly. Their view is that the sizes of some fish are outside of the normal range for the different taxa. Both concerns are significant and will need to be addressed before we could consider publication. However, the referee provides good advice for handling body mass confounds, and also for clarifying the other points of confusion.

Thank you. We are happy that one of the reviewers (#2) approved the revised manuscript, and have revised the manuscript to respond to all the concerns raised by reviewer (#1). A brief outline of our response to these main issues is provided below:

Size and dimensionless numbers:

We applied a size correction to the data, but used a different approach. We attempted to use the size-correction suggested by the reviewer, i.e. to apply the residuals by regressing the non-dimensional numbers on size as input. This would be a straightforward approach, however, on closer examination of the data we found that the relationship between the (log-transformed) non-dimensional numbers and size is not linear, and especially that it deviates from linearity at high and low sizes. This posed a problem for running the subsequent analysis because it meant that species at the extremes had biased size-corrected values.

We opted for a different approach. We size-corrected our data by calculating the speeds (ram speed and gape speed) in units of body length per second, and ran the evolutionary models on this data. This constitutes a bias-free test of whether the evolutionary patterns are caused by body size. As you will see

in the updated ESM file, the results are consistent with the original result, i.e. trophic guilds are associated with the evolution of different size-corrected adaptive peaks.

We include these results in the ESM due to space limitations, but also because this size correction alters the original meaning of our investigation. As you are well-aware, non-dimensional numbers in biophysics have a specific meaning beyond being a ratio of several variables. In our case they have a specific meaning regarding the nature of forces exerted on the body (Re) and mouth cavity (Wom) of the striking fish. We therefore think that it is important to focus the discussion of the results around those meaningful, original variables.

We thus maintain that Re and Wom are appropriate and useful parameters to base this analysis on. This paper combines a phylogenetic approach with biophysics. Here, the non-dimensional Reynolds and Womersley numbers are well understood in the context of hydrodynamics (as outlined in the main text), and provide an order of magnitude analysis on the governing forces occurring during suction-feeding: inertia, viscous, pressure and unsteadiness.

Body size

We appreciate the reviewer's concern regarding accuracy of the fish size as reported. We would like to point out that we obtained our data from previously published studies, i.e. we are reliant on the quality of data in published papers; therefore, we did not have the option to choose the focal fish in the experimental trials. Note that this is a weakness (because data might be noisy) but also a strength, as data collection is agnostic to our hypothesis.

We address the specific comments regarding the size of the fish relative to the assigned feeding guild as follows: 1) We compiled data from previously published papers to test at which size piscivorous fish switch to fish prey. These data indicate that this size is often well below those given for the fish in the data set used herein (see ESM). 2) We note that the possibility that fish are miscategorized constitutes a conservative test of the hypothesis. Hypothetically, assuming the fish used in the studies were, for example, too small to be considered functional piscivores, we would expect no pattern differences between the piscivores and the generalists. This was not the case.

Reviewer(s)' Comments to Author:

Referee: 1

Comments to the Author(s).

This is a revision of a ms I reviewed earlier. The revision is definitely improved but the main issues remain. Body size is correlated with all three factors being investigated, Reynolds number, Womersley number and diet. Although the results are tantalizing, it is not entirely clear that diet shapes the hydrodynamic regime beyond their shared association with scale. There are also some other issues that I want to raise with the new version. These include concerns about the fact that the size of films fish often does not match adult sizes of those fish for which diet came from, and I found some confusing issues in several figures that would need to be addressed.

Thank you. We address these comments as they are presented in detail below.

The core idea in the paper is that the hydrodynamic regime of suction feeding is characteristic of very general feeding guilds – zooplanktivores, piscivores and generalist invertivores. Two non-dimensional numbers are calculated based on videos of 71 species of fish feeding in the lab and these values are evaluated with respect to diet habits – Reynolds number and Womersley number. A discriminant analysis only correctly classifies species into diet category about 50% of the time. Continuous trait model fitting supports the presence of distinct ‘adaptive peaks’ within each dimensionless number for each of the three diet categories. The implication is that feeding on fish prey results in adaptations that result in higher Re and higher Womersley number, while adaptation to lower values of both characterizes planktivores.

The biggest concern offered in my earlier review (and also put forth by the other reviewer) was that Reynolds number, Womersley number and diet can all be expected to be highly correlated with body size. The authors responded to this by showing that their OU model fitting provides a better fit to the two dimensionless numbers than to body size or mouth gape. That is partly satisfying, but I am not convinced by this response. I think the better way to deal with it would be to actually remove the effects of body size from both numbers and to then show that diet regime affects these residuals. The authors did not address the point that diet is associated with size - zooplankton are small and fish prey are large so zooplanktivores tend to be small and piscivores tend to be large. There is a very large literature that shows that both within species (during ontogeny) and across species, piscivory is associated with larger body size (and larger gape). In order to demonstrate adaptation to prey-specific hydrologic regimes, I think you need to remove the effect of body size from the variables that go into these dimensionless numbers to show that size-corrected time to peak gape, ram speed, and gape size combine in a way that shows these different hydrologic regimes independent of body size.

Done, using a different approach: We appreciate the concern that size may constitute a trivial explanation not sufficiently accounted for. We examined the feasibility of the remedy suggested above, i.e. regressing Re and Wom on SL and using the residuals as an input in the evolutionary models. However, we found that the relationship between (log transformed) Re and SL deviates from linearity, especially at high and low values of SL . Thus, the residuals from a linear regression misrepresent observations in these regions of SL and introduce bias to the data.

We therefore attempted to find an alternative approach for size-correction. We recalculated ram and gape speed in terms of body lengths per second, to relativize the numbers (**L239-240, Table SI 2**). This approach removes the effect of size completely and is therefore a critical test of whether the observed patterns are caused just due to size. Interestingly, we again find greater support for multi-peak models patterns, demonstrating that size alone is an insufficient explanation for the differences between feeding guilds. These results are mentioned in the text and included in the ESM (**L307-308, Table SI 2**).

We note that we maintain the discussion of the results concerning the original variables (Re and Wom) in the body of the text. This is because ‘size-corrected’ dimensionless number is, by definition, no longer

dimensionless. While there is a host of literature on the hydrodynamics as pertaining to dimensionless numbers, the meaning of a variable with units $1/m$ is, as far as we are aware, undefined and is also not apparent to us.

We also emphasize in the revised version (**L230-243**) that we do not claim that size has no effect,. However, we make the points that (1): while size affects numerous (ecological, physiological, etc.) we focus on a particular, well defined aspect of the hydrodynamics (which happens to also be affected by size) (**L240-243**), that (2): that a side-by-side comparison of evolutionary models based on the dimensionless numbers and models based on the corresponding size measures (SL and PG) show that size - on its own - is an inferior explanation for the observed trait evolution and (3) that size-corrected speeds also evolve towards guilds-specific peaks, demonstrating again that size is not the only variable driving the evolutionary patterns (**ESM 4.1, Table SI 2, Table SI 4, Fig SI 2, Fig SI 3**).

A new concern is that in this manuscript there is a complex pattern of mismatch between the size of fish upon which determinations of diet were made and the size of specimens used in the video trials. The cichlids are all rather small individuals ranging from 65-95 mm but the centrarchids ranged from 45 to 300 mm and the serranids from 35 to 250. Some of the individuals filmed were radically different from typical adult sizes of the species (which would have been the size fish that diet was determined from). For example the serranid *Variola louti* filmed was 55 mm but a typical adult of this species would be at least 10 times that length. It is not even clear if this species is piscivorous when it is 55 mm – few fish species are piscivores at such a small body size. If the two dimensionless numbers are both strongly affected by body size, then clearly observations on this species should not be taken as representative of adults that are piscivores.

Addressed: We note that both species of *Enneacanthus* and the four *Liopropoma* species (classified as zooplanktivores and generalists) belong to species that have uncharacteristically small size on the background of their clades (centrarchids and serranids, respectively). Their maximal size is <10 cm, and the common size is ~6 cm. This is compared to a maximal size of >20 in the other species of the clade. In this sense, the individuals from these species represent proportional sampling with respect to the size of the specimens, compared to the other species.

As a more general way of validating our classification, we conducted a literature review to obtain information on the body size at which piscivorous fish switch to fish prey. Feeding experiments on the piscivorous walleye (*Sander vitreus*) indicated that fish of 20 mm length selected zooplankton while fish of 40-100 mm selected fish and benthic invertebrates (Galarowicz et al 2006). By comparison, Japanese Spanish mackerel (*Scomberomorus niphonius*) larvae are piscivorous from their first feeding (Shoji et al 2001). Finally, a review by Mittelbach & Persson (1998) on the ontogeny of piscivory shows that the average size at which fish that become piscivorous in their first year is 57.7 mm while the average size at which fish that become piscivorous in their first or second year is 89.7 mm. In our data, out of the 23 piscivorous species, ten were below 89.7 mm, of which eight were small-bodied cichlids, and only one, *Variola louti*, was below 57.7 mm (**ESM 2.2**). With respect to *V. louti*, although no formal study on ontogenetic prey choices was found for this species, advice and information from hobby aquarists

appear to indicate that this is a highly predatory species at any size, which if housed with smaller fish may consume them.

Furthermore, if - hypothetically - the fish used in the feeding trials were too small to be functional piscivores, we would expect no difference between piscivores and generalists, but our analyses reveal a clear difference between the feeding guilds. Thus, the approach we have used here is conservative.

I had not appreciated previously that many of the species values are based on a single sequence for a single species. That is true for all the cichlid species and is worrisome given that kinematics varies from strike to strike and motivation affects ram speed and time to peak gape.

Addressed: Thank you for pointing this out. The data, which also were included for review in the original submission, were obtained from published studies, as described in the Method section. Re-analysis of data obtained from other studies is constrained by how these data were initially collected. However, we have performed a supplemental analysis that excluded cichlids, which were raised as a group of concern, and again found that the same pattern emerged (**Table SI 3, Table SI 4**).

L47. This sentence would need to be rewritten. "There, in which the behaviors..." does not make sense.

Rewritten. Now phrased as "In the aquatic realm, the behaviors of the organisms are..." (**L46**)

L109. acanthomorph, not acantomorph

Corrected. Thank you (**L108**)

L216-230. I find that I am wondering if you have sufficient power with this data set to allow you to use the full multi-peak OU model that allows both sigma and alpha to vary? The results of simulation studies that demonstrate sufficient power should be included in the SI. I confess that it does not really matter as I do think there is sufficient power to distinguish BM, single peak OU and multi-peak OU and that is really what matters. But given all the sources of error in this study that are not being accounted for I don't think you should be taking the different estimates of sigma and alpha very seriously.

Addressed: We are grateful for this advice. Having consulted additional literature on this subject (Si Tung Ho et al. 2014; Cooper et al. 2016; both Methods Ecol Evol), we have removed the most complicated model (OUMVA) from our analyses and all reference to it in the main text, figures and tables. Our focus throughout the analyses has been on the existence of models with no peak, same peak and different peaks (as indicated by the different colours used in **Fig. 3, Table 1, Table SI 2, Table SI 3**). This is the key result in our analyses, and as pointed out, we are confident that this result is solid.

L280. These descriptions of the locations of the adaptive peaks in the OU modeling are not consistent with figure 3, which shows the peak for generalists has the highest Reynolds number and highest Womersley number, not the piscivores as stated here. Indeed the relative position of the three peaks is surprisingly inconsistent in different parts of the paper. For example, I just get completely confused about what is going on when I compare Figure 3, Figure SI 3, and Figure SI 4. I have to say - something seems off here. Maybe there is a mistake in the color coding in some of the figures?

Corrected: Thanks for spotting this! Yes, with the addition of several supplemental analyses (some with different definitions of feeding guilds), the color of this Figure got miscoded. Now corrected. (Figs. 3, SI 3, SI 4)

Figure SI 4. Looking at the plots for cichlids and serranids I am struck by how poor the match of species means are to the appropriate adaptive peaks. In serranids none of the piscivores fall within the 75% confidence lines of the piscivore peak but three species fall within the 75% limits of the generalist peak. The cichlids also seem to be poor matches with the location of the peaks. It really makes it hard to take seriously the locations of the peak when they do not seem to match the empirical data very well.

Addressed: Thanks for alerting us to the need to explain this Figure in greater detail. As we described in the text, the adaptive peak, which is what these models were set up to locate, is the optimal trait value a species is expected to obtain *if allowed to evolve for a sufficiently long time*. But evolution is not always linear, and as we explained when we introduce the phylogenetic modelling concept (L171-181), a zooplanktivorous species that descended from another zooplanktivore is likely to be closer to that peak, compared to a zooplanktivorous species that descended from, for example, a piscivorous ancestor. Deviations between the locations, in trait space, of extant species to the estimated optimal regions, do not invalidate the modelling. Thus we disagree with the criticism, and take the opportunity to clarify the figure in the legend (Fig. SI 4).

Figure SI 1. I am having a hard time seeing how this cluster analysis supports the diet categories. Unless I misunderstand the figure, it shows that piscivores fall out in all four major clusters and that planktivores fall out in three of the four clusters. This would appear to be a very weak match between the three named diet categories and the stomach content data used to make this figure.

Addressed: Thank you for highlighting that this Figure was insufficiently explained. We interpret the figure as displaying three levels of organization; diet, habitat, and family, though the latter two overlap as two families are exclusively freshwater (cichlids and centrarchids) and the other three are exclusively marine (antennarids, pomacentrids, and serranids). We addressed this point with respect to the effect of a freshwater and marine habitat on food items, and showed additional analyses to support our overall conclusion. We have rewritten the text to clarify this (ESM Section 2.1). We have also redrawn Fig SI 1: the analysis is the same but we have utilized the property of dendrograms that allow us to rotate clades about their (vertical) axes, and we have labelled the major clusters to emphasize the pattern (ESM Fig. SI 1).

Referee: 2

Comments to the Author(s).

I am satisfied that the authors have addressed my concerns.

Thank you!